# An artificial intelligence accelerated virtual screening platform for drug discovery

Guangfeng Zhou [1,2,10], Domnita-Valeria Rusnac[3,10], Hahnbeom Park [4,5], Daniele Canzani[6], Hai Minh Nguyen[7], Lance Stewart [2], Matthew F. Bush [6], Phuong Tran Nguyen[8], Heike Wulff [7], Vladimir Yarov-Yarovoy [8,9], Ning Zheng [3] ✉ & Frank DiMaio [1,2] ✉

Structure-based virtual screening is a key tool in early drug discovery, with growing interest in the screening of multi-billion chemical compound libraries. However, the success of virtual screening crucially depends on the accuracy of the binding pose and binding affinity predicted by computational docking. Here we develop a highly accurate structure-based virtual screen method, RosettaVS, for predicting docking poses and binding affinities. Our approach outperforms other state-of-the-art methods on a wide range of benchmarks, partially due to our ability to model receptor flexibility. We incorporate this into a new open-source artificial intelligence accelerated virtual screening platform for drug discovery. Using this platform, we screen multi-billion compound libraries against two unrelated targets, a ubiquitin ligase target KLHDC2 and the human voltage-gated sodium channel Na$_V$1.7. For both targets, we discover hit compounds, including seven hits (14% hit rate) to KLHDC2 and four hits (44% hit rate) to Na$_V$1.7, all with single digit micromolar binding affinities. Screening in both cases is completed in less than seven days. Finally, a high resolution X-ray crystallographic structure validates the predicted docking pose for the KLHDC2 ligand complex, demonstrating the effectiveness of our method in lead discovery.

Structure-based virtual screening plays a key role in drug discovery by identifying promising compounds for further development and refinement. With the advent of readily accessible chemical libraries with billions of compounds[1], there has been an increasing interest in screening the expansive chemical space for lead discovery. Despite the benefits of screening these ultra-large libraries[2], only a few successful virtual screening campaigns using ultra-large libraries have been reported[3]. Moreover, virtual screening of an entire ultra-large library becomes increasingly time and cost prohibitive for physics-based docking methods. In recent years, a number of techniques have been introduced to accomplish the ultra-large library virtual screening, including the development of scalable virtual screening platforms to parallelize docking runs on high-performance computing clusters (HPC)[4], deep learning guided chemical space exploration or active learning techniques to only screening a small portion of the library for similar performance[5–8], hierarchical

[1]Department of Biochemistry, University of Washington, Seattle, WA, USA. [2]Institute for Protein Design, University of Washington, Seattle, WA, USA. [3]Howard Hughes Medical Institute, Department of Pharmacology, University of Washington, Seattle, WA, USA. [4]Brain Science Institute, Korea Institute of Science and Technology, Seoul, Republic of Korea. [5]KIST-SKKU Brain Research Center, SKKU Institute for Convergence, Sungkyunkwan University, Suwon, Republic of Korea. [6]Department of Chemistry, University of Washington, Seattle, WA, USA. [7]Department of Pharmacology, University of California Davis, Davis, CA, USA. [8]Department of Physiology and Membrane Biology, University of California Davis, Davis, CA, USA. [9]Department of Anesthesiology and Pain Medicine, University of California Davis, Sacramento, CA, USA. [10]These authors contributed equally: Guangfeng Zhou, Domnita-Valeria Rusnac. ✉e-mail: nzheng@uw.edu; dimaio@u.washington.edu

structure-based virtual screening[9], and GPU accelerated ligand docking[10].

However, the success of the virtual screening campaigns using the aforementioned techniques depends crucially on the accuracy of the ligand docking programs used to predict the protein-ligand complex structure as well as to distinguish and prioritize the true binders from non-binders. Leading physics-based ligand docking programs, such as Schrödinger Glide[11–13], CCDC GOLD[14], along with their virtual screening platforms for ultra-large library screens, are not freely available to researchers. Autodock Vina[15], as one of the widely used free programs, has slightly lower virtual screening accuracy compared to Glide. Moreover, there is a lack of an open-source scalable virtual screening platform that employs active learning for ultra-large chemical library virtual screens. The emergence of deep learning technology has led to a number of models[16–20] aimed at predicting protein-ligand complex structure in a significantly reduced time. However, these methods are better suited for blind docking problems, where the binding site of the small molecule is unknown. In scenarios where the binding site is known, which is often the case in virtual screening, physics-based ligand docking methods continue to outperform deep learning models[21]. In addition, the deep learning methods are less generalizable to unseen complexes[22].

In this work, we aim to develop a "state-of-the-art" (SOTA), physics-based virtual screening method and an open-source virtual screening platform capable of robustly and efficiently screening multi-billion chemical compound libraries. This is achieved by improving our prior physics-based Rosetta general force field (RosettaGenFF)[23] for virtual screening, yielding an improved forcefield named RosettaGenFF-VS. Based on this new force field, we develop a state-of-the-art virtual screening protocol using Rosetta GALigandDock[23], (hereafter referred to as RosettaVS). In addition, we adopt a docking protocol from our previous work to allow for substantial receptor flexibility, enabling us to model flexible sidechains as well as limited backbone movement in our virtual screening protocol. This proves critical for certain targets that require the modeling of induced conformational changes upon ligand binding. We then create a highly scalable, open-source AI accelerated virtual screening platform (OpenVS) platform integrated with all necessary components for drug discovery (Fig. 1a). We use the OpenVS platform to screen multi-billion chemical compound libraries against two unrelated proteins: KLHDC2[24,25], a human ubiquitin ligase and the human voltage-gated sodium channel $Na_V1.7$[26]. The whole virtual screening process is finished within seven days on a local HPC cluster equipped with 3000 CPUs and one RTX2080 GPU for each target. From the initial virtual screening campaigns, we discover one compound for KLHDC2 and four compounds for $Na_V1.7$, all exhibiting single-digit µM binding affinity. Using a focused library with our virtual screening platform leads to the discovery of six more compounds with similar binding affinities to KLHDC2. Finally, the docked structure of the KLHDC2 complex is validated by X-ray crystallography, showing remarkable agreement

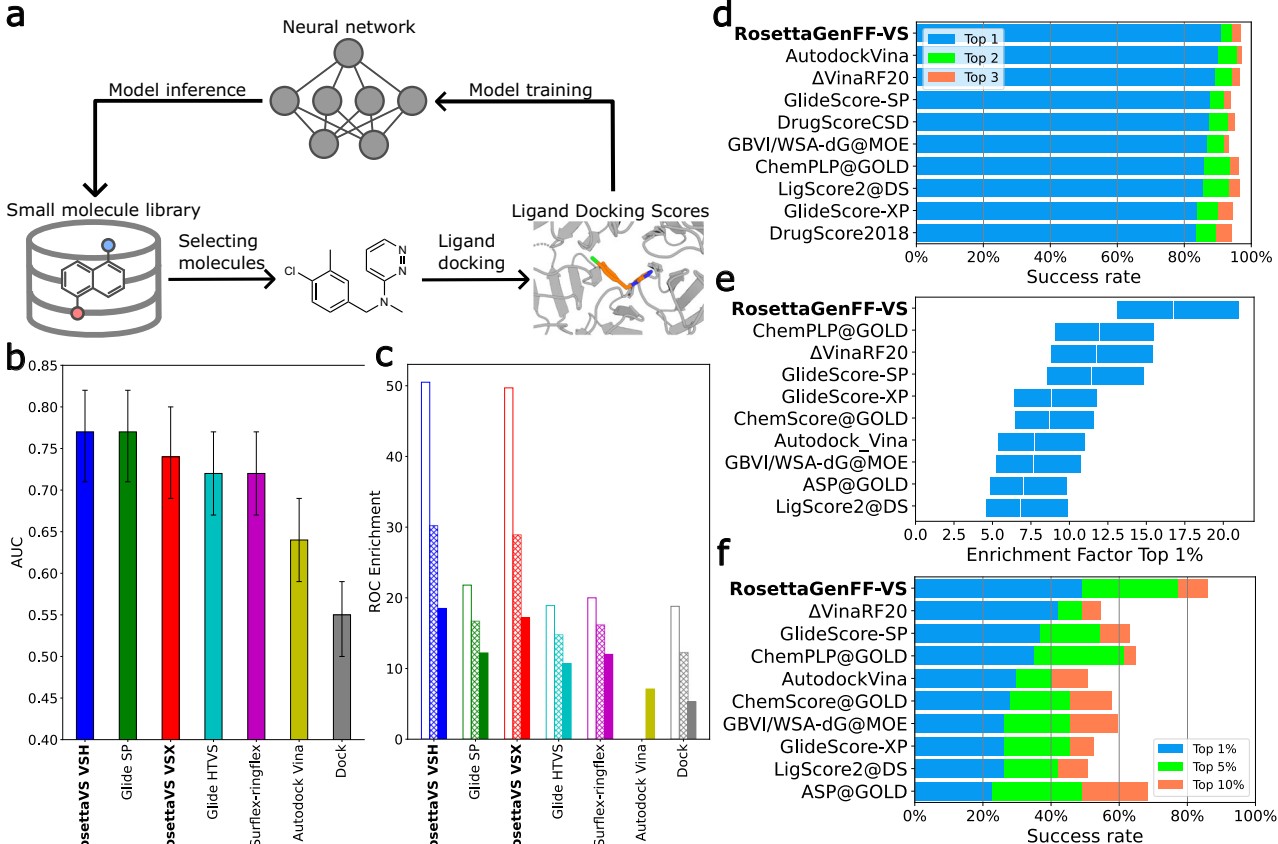

**Fig. 1 | Deep learning guided virtual screening approach and the state-of-the-art ligand docking method. a** Overview of the deep learning guided virtual screening protocol. **b** Results of the area-under-curve (AUC) of the receiver operator characteristics (ROC) curve of the DUD benchmark, averaged over three independent runs ($n = 3$) and averaged over forty targets ($n = 40$), 95% confidence intervals are shown as error bars. **c** Mean ROC enrichments of the DUD benchmark at 0.5% (empty bar), 1% (patterned bar), and 2% (solid bar) false positive rates, results are averaged over three independent runs ($n = 3$) and averaged over forty targets ($n = 40$). Results of other methods are obtained from Ref. 80–82, the same color scheme is used as in subpanel (**b**). **d** CASF2016 docking power, the docking success rates of the top ten methods are shown. **e** CASF2016 screening power, the top 1% enrichment factors with a 90% confidence interval of the top ten methods are shown. **f** CASF2016 screening power, the success rates of the top ten methods are shown. The CASF2016 docking power and screening power results of all the methods can be found in Supplementary Fig. 3–5. Results of other methods are obtained from Ref. 28. Source data are provided as a Source Data file.

with the predicted binding pose. This iterative process of exploration, curation, and testing underscores the robustness of our methodology and its potential for uncovering promising compounds in large molecular libraries.

## Results

### Development of an AI-accelerated virtual screening platform

Our previously developed Rosetta GALigandDock is a ligand docking method that uses a physics-based force field, RosettaGenFF, that previously has shown superior performance in ligand docking accuracy[23]. This method allows for the accurate modeling of protein-ligand complexes, accommodating full flexibility of receptor side chains and partial flexibility of the backbone. However, it is not directly applicable for large-scale virtual screening due to: (a) its inability to accurately model certain functional groups (as the original method was tested on hundreds of thousands of compounds rather than the billions in this study); and (b) its lack of an entropy model for accurate ranking of different compounds binding to the same target. Moreover, it is prohibitively expensive to dock each individual compound in a multibillion chemical compound library using physics-based virtual screening methods.

To address these issues, we incorporated several enhancements and rectified several critical issues to facilitate the modeling of billions of small molecules. Firstly, we have improved RosettaGenFF by incorporating new atom types and new torsional potentials and improved the preprocessing script (see "Methods"). Second, we have developed RosettaGenFF-VS for virtual screening to rank different ligands binding to the same target, which combines our previous model's enthalpy calculations ($\Delta H$) with a new model estimating entropy changes ($\Delta S$) upon ligand binding (see "Methods" for details).

To enable screening against ultra-large compound libraries, we employed two strategies. First, we developed a modified docking protocol, RosettaVS, that implements two high-speed ligand docking modes: virtual screening express (VSX) is designed for rapid initial screening, while the virtual screening high-precision (VSH) is a more accurate method used for final ranking of the top hits from the initial screen. The key difference between the two modes is the inclusion of full receptor flexibility in VSH (see "Methods" for further details).

Even with these speedups, docking more than one billion compounds is prohibitively expensive. Building upon recent works[4–8], we developed an open-source virtual screening (OpenVS) platform that uses active learning techniques to simultaneously train a target-specific neural network during the docking computations to efficiently triage and select the most promising compounds for expensive docking calculations. This platform was also designed to be highly scalable and parallelizable for large-scale virtual screens.

### RosettaVS shows state-of-the-art performance on virtual screening benchmarks

We first used the Comparative Assessment of Scoring Functions 2016 (CASF2016) dataset[27,28] to benchmark the performance of RosettaGenFF-VS. The CASF2016, consisting of 285 diverse protein-ligand complexes, is a standard benchmark specifically designed for scoring function evaluation. It provides all small molecule structures as decoys, effectively decoupling the scoring process from the conformational sampling process inherent in molecular docking. We used the docking power test to benchmark the docking accuracy and the screening power to benchmark the screening accuracy. Recent developments of deep learning based score functions have demonstrated superior performance on these benchmarks[16,20,29], however, it is not clear how generalizable these methods are to unseen compounds and receptors. Moreover, these methods have not employed stringent train/test splits. Even when a cutoff of 0.6 Tanimoto similarity for ligands and a sequence identity of 30% for proteins were used, it is likely that the contamination of these validation benchmarks

still occurred. Because of this, our subsequent comparisons will focus on other physics-based scoring functions, including the top-performing ones from ref. 28. As shown in Fig. 1d and Supplementary Fig. 3, RosettaGenFF-VS achieves the leading performance to accurately distinguish the native binding pose from decoy structures. Further analysis of binding funnels, which measures the efficiency of the energy potential in driving the conformational sampling toward the lowest energy minimum, shows RosettaGenFF-VS's superior performance across a broad range of ligand RMSDs, suggesting a more efficient search for the lowest energy minimum compared to other methods (Supplementary Fig. 7). Next, the screening power test was conducted to assess the capability of a scoring function to identify true binders among a multitude of negative small molecules. Two metrics are used to assess the performance of scoring functions in the screening power test. The first metric is the enrichment factor (EF) which measures the ability of the docking calculations to identify early enrichment of true positives (see "methods" for details) at a given X% cutoff of all the compounds recovered. The second metric is the success rate of placing the best binder among the 1%, 5%, or 10% top-ranked ligands overall target proteins in the dataset. In Fig. 1e and Supplementary Fig. 4, the top 1% enrichment factor from RosettaGenFF-VS ($EF_{1\%} = 16.72$) outperforms the second-best method ($EF_{1\%} = 11.9$) by a significant margin. Similarly, Fig. 1f and Supplementary Fig. 5 illustrate that RosettaGenFF-VS excels in identifying the best binding small molecule within the top 1/5/10% ranking molecules, surpassing all other methods. Analysis of our method on various screening power subsets[28] shows significant improvements in more polar, shallower, and smaller protein pockets compared to other methods (Supplementary Fig. 8). However, in a realistic virtual screening scenario, the docking method must accurately score the complex while also effectively sampling the conformations.

To this end, we further evaluated the virtual screening performance of VSX and VSH mode from RosettaVS protocol on the Directory of Useful Decoys (DUD) dataset[30]. The DUD dataset consists of 40 pharmaceutical-relevant protein targets with over 100,000 small molecules. Two common metrics, AUC and ROC enrichment, are used to quantify the virtual screening performance. The receiver operating characteristic (ROC) curve has been widely used to evaluate virtual screening performance where the aim is to distinguish between active and decoy compounds. The area under the ROC curve (AUC) assesses the overall performance of a method to differentiate actives vs decoys. ROC enrichment, which addresses a few deficiencies of the enrichment factor[31], measures the true positive enrichment at a given X% false positive rate. The early enrichment is a critical factor in large-scale virtual screens, as the limitations of current experimental throughput typically allow for the synthesis and experimental testing of only dozens to hundreds of compounds. The results, in terms of AUC and ROC enrichment, position RosettaVS as the leading virtual screening method (Fig. 1b, c and Supplementary Fig. 9). Notably, RosettaVS outperforms the second-best method by a factor of two (0.5/1.0% ROC enrichment), achieving state-of-the-art performance in early ROC enrichment, further highlighting the effectiveness of RosettaVS. Furthermore, VSH mode slightly outperforms VSX mode due to the capability of modeling the conformational changes of the pocket sidechains induced by the ligand(see "Methods").

### Discovery of small molecule hits to KLHDC2 ubiquitin ligase

In order to showcase the effectiveness of our newly developed method, we embarked on a large-scale virtual screening against the human KLHDC2 ubiquitin ligase[24,25], which has not yet been linked with any known drug-like small molecule binder. As a substrate receptor subunit of the CUL2-RBX1 E3 complex, KLHDC2 features a KELCH-repeat propeller domain, which can recognize the di-glycine C-end degron of its substrates with a nanomolar affinity. We set out to identify compounds that can anchor to the diglycine-binding site of

KLHDC2, which has recently been suggested as a promising PROTACs E3 platform for targeted protein degradation[32,33].

We used the OpenVS platform and VSX mode in RosettaVS to screen the Enamine-REAL library against the target protein structure, which contains ~5.5 billion purchasable small molecules with an 80% synthesis success rate (see "Methods"-Workflow of AI accelerated virtual screening). As shown in Supplementary Fig. 11, the protocol discovers better compounds with higher predicted binding affinity after each docking iteration. The predicted binding affinity for the top 0.1 percentile significantly improved from −6.81 kcal/mol in the first iteration to −12.43 kcal/mol in the final iteration. We stopped the virtual screening when it reached the maximum tenth iteration, especially since no new global minimum structures were identified beyond the eighth iteration. Subsequently, we re-docked the top-ranked 50,000 small molecules from the virtual screening using VSH mode in RosettaVS, allowing for flexibility in the receptor structure during docking. The entire computation was completed within a week on a local HPC cluster equipped with 3000 CPUs and one RTX2080 GPU. Approximately 6 million compounds (0.11%) from the Enamine REAL library were subjected to docking.

We took the top-ranked 1000 compounds and filtered out compounds with low predicted solubility, unsatisfied hydrogen bonds in the bound conformation, and followed by similarity clustering to reduce the redundancy in ligand structures. A total of 54 molecules that passed the filtering and clustering were manually examined for favorable interactions and geometries in PyMol[34]. Finally, 37 compounds were chosen for chemical synthesis. Out of these, 29 compounds were successfully synthesized (Supplementary Fig. 13) and characterized in an AlphaLISA competition assay, in which each compound was tested for its ability to compete with a diglycine-containing SelK C-end degron peptide for binding KLHDC2. While several compounds showed detectable activity in displacing the degron peptide, compound 29 (C29) stood out with the best IC$_{50}$ of ~3 μM (Fig. 2a, c). This single digit μM IC$_{50}$ was further confirmed in a competition assay using BioLayer Interferometry (Supplementary Fig. 14).

To reveal the binding mode of compound 29, we soaked the crystals of apo KLHDC2 with the compound and determined the structure of the KLHDC2-C29 complex at 2.0 Å resolution[35]. Consistent with its activity in displacing the diglycine peptide, compound 29 binds to the degron-binding pocket with its distal carboxyl group interacting with two critical arginine residues (Arg236 and Arg241) and a serine residue (Ser269) in KLHDC2 that are involved in recognizing the extreme C-terminus of the degron[24] (Fig. 3). The triazole moiety next to the carboxyl group of the compound is nestled among three aromatic residues (Tyr163, Trp191, and Trp270) and stabilized by a NH...N hydrogen bond. The interaction of the compound with the E3 is further strengthened by a hydrogen bond formed between Lys147 and the central carbonyl group of the small molecule as well as direct packing of the tert-butylphenyl group to the auxiliary chamber of the degron-binding pocket[24]. In contrast to the two ends of the compound, the dimethyl-sulfide linker in the middle of the compound shows poor electron density, indicating higher structural flexibility (Fig. 2e). Overall, the binding mode of compound 29 is highly similar to that of the diglycine C-end degron with a binding pose closely matching the prediction (Fig. 2f).

Following our initial hit, we broadened our exploration to the ZINC22 library[36], which houses approximately 4.1 billion small molecules in ready-to-dock 3D format. It's noteworthy that a substantial fraction of ZINC22 originates from the Enamine REAL library. We performed a substructure search of the acetyl-amino-methyl-triazole-acetic acid backbone (2D structure highlighted in red in Fig. 2a, b) against the ZINC22 library and identified ~381,567 compounds. These compounds are docked using GALigandDock flexible docking mode, and 21 compounds were picked by manually examining the top 100

structures that passed all the filters mentioned above. These compounds were synthesized (Supplementary Fig. 15), and their activities were tested in the AlphaLISA-based competition assay with compound C29 as a positive control. Remarkably, six additional hits showed single digit μM IC$_{50}$, further validating the effectiveness of our method (Fig. 2b, d). Future optimization will be needed to improve the potency of these compounds to reach the nanomolar range. To test the reliability of our screening procedure, we reran a portion of our computational experiment and rediscovered the confirmed best-hit compound C2.8. (see "Methods" for details)

## Discovery of small molecule antagonists to Na$_V$1.7 VSD4

To evaluate the wider applicability of our virtual screening protocol, we examined its effectiveness on the human voltage-gated sodium channel, hNa$_V$1.7. Specifically, we targeted voltage-sensing domain IV (VSD4), which is involved in Na$_V$ channel fast inactivation[37–40] and contains a receptor site for small molecules that stabilize an inactivated state of the channel[26,41,42]. We used the same virtual screening protocol to screen the target against the ZINC22 library (~4.1 billion compounds). Similar to the KLHDC2 screen, new compounds with better predicted binding affinities were discovered after each iteration and the predicted binding affinity for the top 0.1 percentile improved from −10.8 kcal/mol in the first iteration to −18.2 kcal/mol in the final iteration. The virtual screening was stopped after the seventh iteration, where the top predicted binding affinities reached convergence (Supplementary Fig. 12). The top-ranked 100,000 small molecules from the virtual screening were re-docked using VSH mode in RosettaVS to account for the flexibility in the receptor structure. Approximately 4.5 million compounds (0.11%) from the ZINC22 library were subjected to docking.

We first clustered the top 100,000 ranked small molecules, then applied filters on the top 1000 cluster representative molecules. A total of 160 molecules that passed the clustering and filtering were examined manually. To ensure the chemical novelty of our selection, we specifically excluded molecules that contain the known arylsulfonamide warhead or structurally resemble antihistamines or beta-receptor blockers. Finally, ten molecules with Tanimoto similarities of less than 0.33 to the known inhibitors of Na$_V$1.7 from the ChEMBL database[43,44] were selected for synthesis. Of these, nine were successfully synthesized (Supplementary Fig. 16), and their activities were measured using the whole-cell patch-clamp electrophysiology assay on hNa$_V$1.7 channel stably expressed in HEK-293 cells as described in Methods. Compound Z8739902234 demonstrated the highest potency with IC$_{50}$ = 1.3 μM for Na$_V$1.7 in an inactivated state-dependent manner (Fig. 4 and Supplementary Fig. 18). IC$_{50}$ values of better than 10 μM were observed for four compounds, translating to a hit rate of 44.4% (Supplementary Fig. 17). Notably, compound Z8739902234 is state dependent (Supplementary Fig. 18, left panel) and has moderate selectivity for hNa$_V$1.7 versus hNa$_V$1.5 and hERG channels (Supplementary Fig. 18, right panel).

## Discussion

In this work, we presented a state-of-the-art physics-based virtual screening method integrated into a comprehensive scalable platform that uses active learning for large-scale virtual screens and lead discovery. Our approach led to the discovery of seven binders to a new E3 ligase, KLHDC2, and four binders to the human voltage-gated sodium channel Na$_V$1.7 VSD4.

The superior performance of RosettaGenFF-VS and RosettaVS on the CASF2016 and DUD benchmarks, respectively, establishes it as a leading physics-based method for ligand docking and virtual screening. The notable performance of RosettaVS comes from two major advances. Firstly, the combination of high docking accuracy and sampling efficiency allows the virtual screening protocol equipped with RosettaGenFF-VS to find the correct binding minimum of the

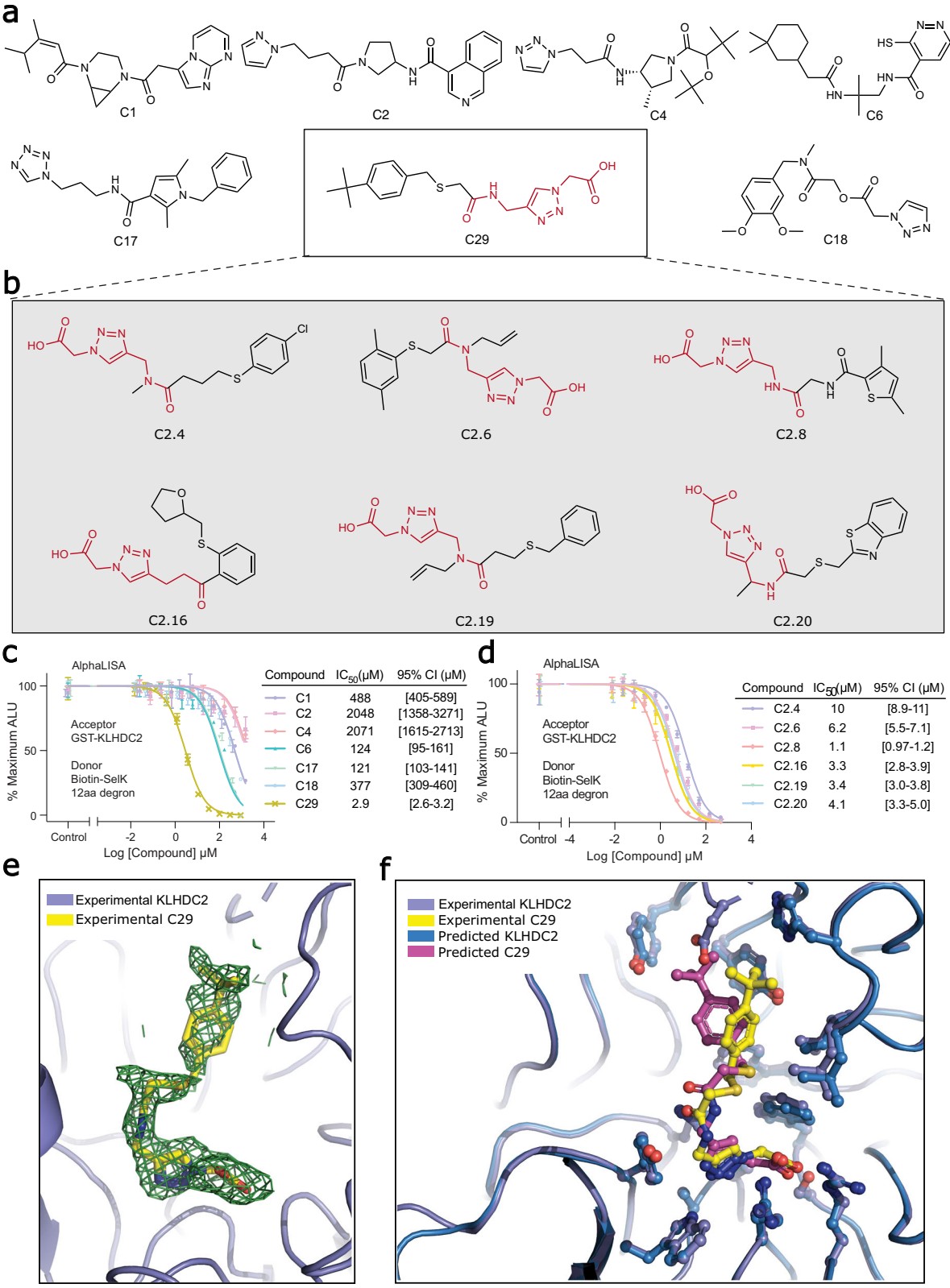

protein-ligand complex more effectively than other methods. Secondly, unlike most other virtual screening methods that tend to work well only on more hydrophobic, deeper, and larger protein pockets, our method also demonstrates high-performance with more polar, shallower, and smaller pockets, likely due to the better balance of protein-ligand versus intra-ligand molecular energies achieved by RosettaGenFF-VS.

Although our methodology outperforms existing approaches in all aspects, we believe there is room for further improvement. The surge in the application of artificial intelligence across various scientific domains, including protein structure prediction[45,46], drug discovery[47,48], and materials design[49], has been a notable trend in recent years. Future enhancements to our protocol will likely involve the integration of GPU acceleration and deep learning models, e.g.,

**Fig. 2 | Deep learning accelerated virtual screening finds KLHDC2 binders. a** 7 out of 29 initial synthesized and assayed compounds from the initial virtual screening. **b** 6 out of 21 synthesized and assayed compounds from the focused screening. Seven compounds in total show low micromolar binding affinity (indicated by the boxes). The substructure highlighted in red is used for focused library generation. **c** AlphaLISA assay and the $IC_{50}$ values of the seven compounds from the initial screening. The statistics were calculated based on three technical replicates ($n$ = 3) for each concentration of the compounds in the 12-point titration curves. Data are presented as mean values +/− SD. **d** AlphaLISA assay and the $IC_{50}$ values of

the six compounds from the focused screening. The statistics were calculated based on three technical replicates ($n$ = 3) for each concentration of the compounds in the 12-point titration curves. Data are presented as mean values +/− SD. **e** Close-up view of C29 bound to KLHDC2 together with its $mF_o$-$DF_c$ map calculated before the compound was built into the model and contoured at 1.5 σ. **f** Comparison of experimentally resolved and the computationally predicted binding pose of C29. The high-resolution X-ray crystal structure in yellow is superimposed on predicted docked pose in magenta.

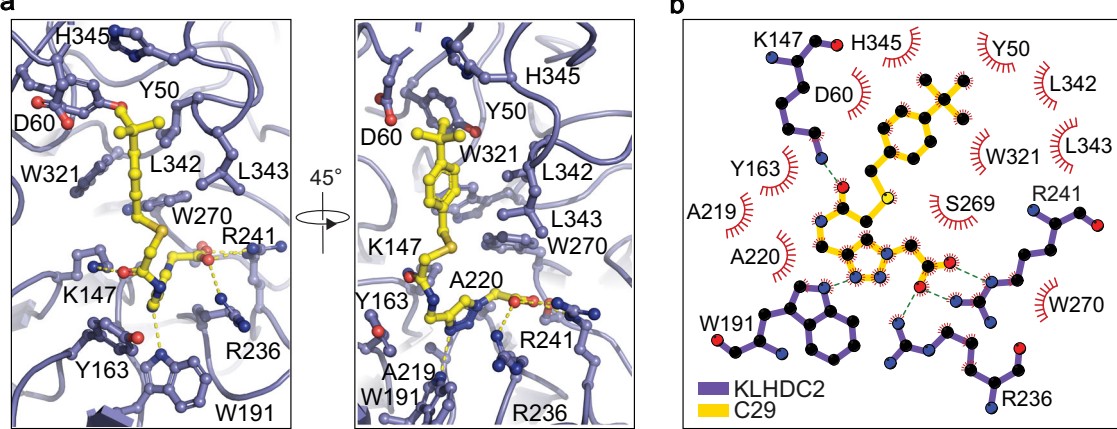

**Fig. 3 | Crystal structure of KLHDC2-C29 complex. a** Specific residues (blue) interacting with C29 (yellow) are labeled and shown as sticks and balls. Hydrogen bonds are shown as yellow dashed lines. **b** 2D representation of interactions between C29 (yellow) and KLHDC2 (blue). Hydrogen bonds are shown as blue dashed lines. Residues in the pocket that form hydrogen bonds with C29 are shown as sticks and balls.

using GPU to accelerate ligand docking[10] or using generative AI for efficient pose generation[50,51]. Other potential improvements include refining the surrogate active learning models for better guidance of chemical-space exploration and incorporating a generalizable deep learning-based score function for improved discrimination of true binders. Another improvement to our approach is to enable the use of known non-small molecule binders, such as macrocycles or antibody loops, as template structures to guide small molecule virtual screening. We anticipate that further developments of structure-based virtual screening combined with deep learning techniques will significantly improve the accuracy and efficiency of virtual screening campaigns.

## Methods

### Computational methods

The computational methods are organized into three primary sections. The first section focuses on the development of the Rosetta general forcefield for virtual screening (RosettaGenFF-VS) and the Rosetta virtual screening protocol (RosettaVS). The second section presents the benchmarks of RosettaGenFF-VS and RosettaVS. The final section provides detailed information about the AI-accelerated virtual screening protocol.

### Development of RosettaGenFF-VS and RosettaVS

In this section, we present detailed information about the developments of the entropy models used to augment Rosetta's general forcefield to enable the ranking of different ligands binding to the same target and the Rosetta virtual screening protocols we developed within Rosetta GALigandDock.

**Entropy estimation.** In a virtual screening task, the entropic contribution to binding free energy caused by "freezing" ligand torsions and rigid-body DOFs upon binding is approximated. The contribution from the receptor is not considered. The entropy change upon binding

takes the following form:

$$\Delta S = w_1 \Delta S_{rb} + w_2 \Delta S_{torsion} \tag{1}$$

The entropy change from ligand rigid body degrees of freedom ($\Delta S_{rb}$) is approximated as a function of molecular mass ($m$)[52]. The entropy change from ligand torsions ($\Delta S_{torsion}$) is directly estimated from its entropy in the unbound state, assuming that absolute entropy in the bound state is negligible. To estimate the probability distribution of each rotatable torsion value at an unbound state, conformations from every 3000 steps of ligand-only Monte Carlo (MC) simulation using RosettaGenFF at room temperature (300 K) are collected. Then the probability distribution is converted into entropy as:

$$\Delta S_{torsion} = RT \sum_i \sum_j (-p_{ij} \log p_{ij}) \tag{2}$$

where R is the gas constant, T is 300 K, and $p_{ij}$ is the probability of a given torsion angle i being in bin j (with 60° bin size). Because we assume ligand torsions are independent, the net entropy loss from ligand torsions is simply the sum of these factors of overall rotatable torsions. This scheme effectively captures the pre-organization effects of ligand torsion angles at their unbound state, therefore addressing shortcomings in simpler algorithms that treat these angles as fully free when unbound. The optimal weights for $\Delta S_{rb}$ and $\Delta S_{torsion}$ were obtained using a grid search to maximize the AUROC of a non-overlapping subset of DUD-E set[53] (see SI, 'Subset of DUD-E' for more details). The weights (w1, w2) considered in this search were {(0,0), (0,1), (1,0), (1,1), (0,2), (2,0), (2,1), (1,2), (2,2)}. The optimal weights obtained (2.0 and 1.0 for rigid-body and torsion angles, respectively) did not vary much from a naive guess of uniform weights (1.0 for both). Compared to the uncorrected results, including entropy estimation improved the AUROC metric by 3% and had a negligible effect on

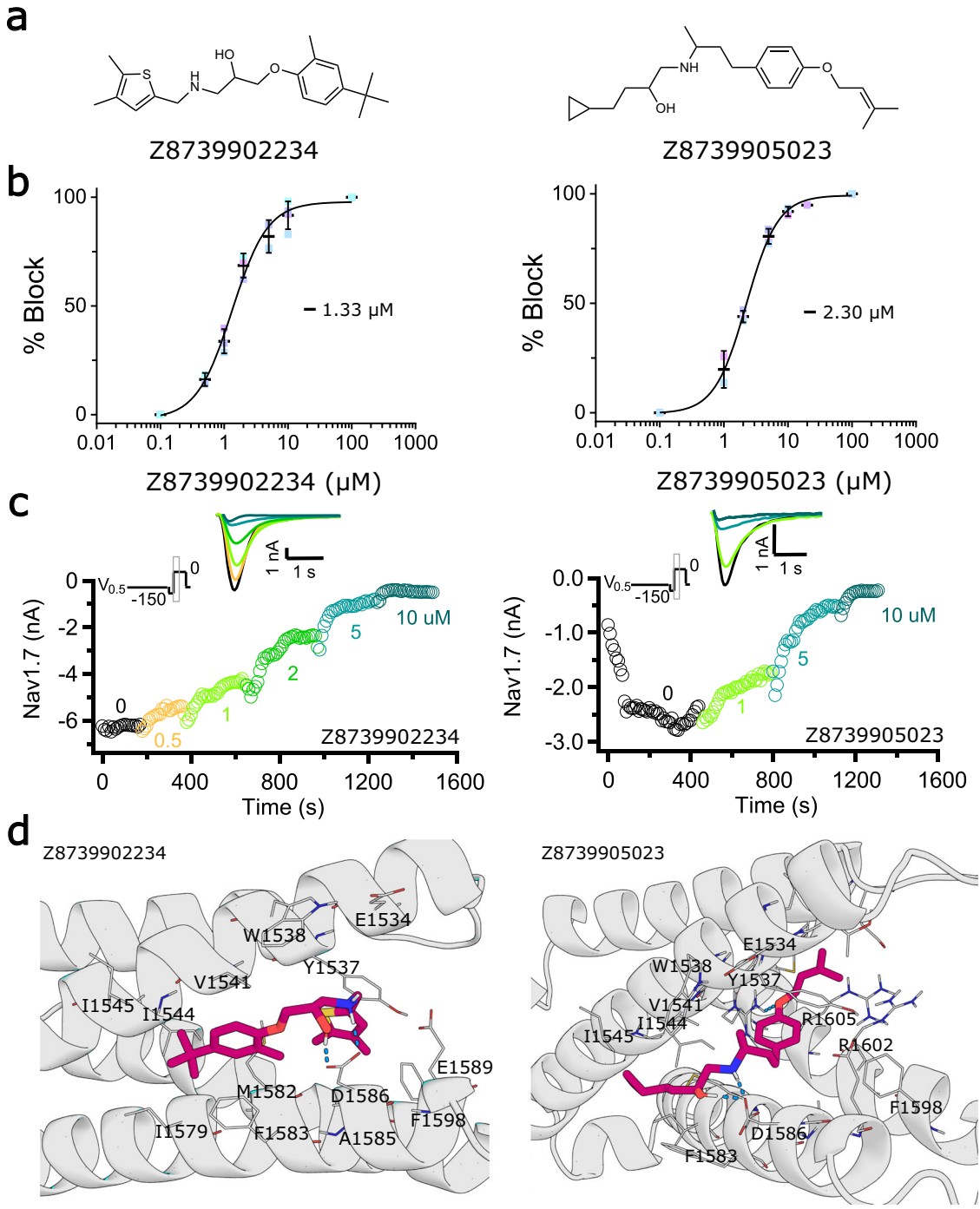

**Fig. 4 | Deep learning accelerated virtual screening finds Nav1.7 binders. a** 2D structures of the best two compounds discovered from the initial virtual screening. **b** Concentration-response curves and inactivated-state IC$_{50}$ values (in μM, mean, 95% CI) for Z8739902234 (1.33, 1.14−1.55) and Z8739905023 (2.30, 2.14−2.46). **c** Exemplary current traces show that Z8739902234 and Z8739905023 inhibit the inactivated state of Na$_V$1.7. **d** Docked structure of Z8739902234 and Z8739905023. Ligands are shown in dark magenta, and human Na$_V$1.7 - NavAb channel chimera VSD4 is shown in gray. Pocket residues that are within 4 Å of the ligand are shown as lines and are labeled. Hydrogen bonds are shown as blue dashed lines.

enrichment-based metrics. This is the default entropy estimation method, thus we termed it "Default Entropy".

A second entropy estimation method was also implemented. This method, which we have named "Simple Entropy", assumes a simpler formulation compared to the first method. Instead of utilizing MC simulation, we used the number of rotational bonds to estimate the torsional entropy of the unbound small molecule. Although this method overlooks the pre-organization effect of the ligand in its

unbound state, we observed comparable benchmark results using this simpler approach. The equation for this simpler approach is as follows:

$$\Delta S = w_1 \Delta S_{rb} + w_2 \Delta S_{torsion} = w_1 \log(m) + w_2 n_{rotor} \tag{3}$$

The optimal weights were determined using a grid search ranging from 0.0 to 3.0, with a step size of 0.1. This was done to maximize the correlation between the predictions and experimental binding

 

affinities derived from a dataset curated from the PDBbind[54] refine dataset. This curated dataset termed the "PDBbind-refine-no-metal" dataset, was prepared by excluding cases that had metal ions present in the complex. To prevent data leakage, any case in CASF2016 was also removed from the "PDBbind-refine-no-metal" dataset. The optimal weights were found to be $w_1 = 0.0$ and $w_2 = 0.4$, indicating that molecular weight does not contribute to the entropy estimation. Therefore, the final formulation of the "Simple Entropy" is $\Delta S = 0.4 * n_{rotor}$.

**Binding affinity estimation.** Binding affinity is estimated by adding together the enthalpic and entropic contributions. The enthalpy changes ($\Delta H$) upon binding is estimated using the following equation:

$$\Delta H = E_{complex} - E_{protein} - E_{ligand} \qquad (4)$$

where $E_{complex}$, $E_{protein}$, $E_{ligand}$ are the Rosetta energy of the complex, the protein, and the ligand, respectively, from the Rosetta general force field (RosettaGenFF). RosettaGenFF combines the Rosetta protein energy model[55–57] and generic energy terms for non-protein molecules, i.e., Lennard-Jones, Coulomb, hydrogen-bond, implicit-solvation and generic torsion energy. A complete description of RosettaGenFF can be found in ref. 23 $\Delta H$ is essentially the interaction energy from RosettaGenFF between the protein and the ligand. The predicted binding affinity is then calculated using the equation: $\Delta G = dH - TdS = \Delta H + \Delta S$, where temperature T is implicitly included in $\Delta S$ estimation described above. This estimated binding affinity is used for ranking different ligands that bind to the same target. It's worth noting that the scale of the estimated binding affinity is arbitrary, and it doesn't directly correspond to the experimental binding affinity. We have named this entropy-augmented force field as the Rosetta general force field for virtual screening (RosettaGenFF-VS).

**Rosetta virtual screening (RosettaVS) evaluation mode.** The evaluation mode in RosettaVS was developed for binding affinity estimation of complex structures. It offers several options for estimating the binding affinity. It can estimate the binding affinity for the provided structure, perform local minimization of the ligand alone within the pocket, or carry out local minimization of the ligand and a set of pocket residues within a certain cutoff (default is 4.5 Å) of the ligand. The minimization can be performed with or without coordinate constraints. These options allow for a more accurate and tailored approach to estimate binding affinities without docking the ligands.

**Rosetta virtual screening (RosettaVS) screening mode.** RosettaVS leverages the "run mode" in our previously developed GALigandDock for various tailored tasks. In this work, we have refined and introduced two run modes for fast ligand docking with binding affinity estimation using RosettaGenFF-VS. These are the Virtual Screening eXpress (VSX) and Virtual Screening High Precision (VSH). The VSX mode treats protein side chains as rigid and executes five iterations with a gene pool size of 50. This mode is designed for quick and efficient screening. On the other hand, the VSH mode allows for flexible pocket residue side chains during docking and conducts a two-stage conformational search on the precomputed energy grid. The first stage up-weighs the coulombic interactions threefold and runs for five iterations with a gene pool size of 100. The second stage uses the default weight for coulombic interactions and also runs for five iterations with a gene pool size of 100. In terms of runtime, the VSX mode takes ~90 to 150 seconds to screen a ligand, while the VSH mode runs about six times slower. Despite the difference in speed, both modes support docking multiple ligands in a single batch. This feature significantly reduces the load on the file system by reading in a single input file for multiple ligands and outputting a single output file for multiple ligands. The choice of the batch size can be arbitrary to best utilize the

local computing cluster. For instance, we used a batch size of 50 for the VSX mode and 5 for the VSH in this study. Detailed settings for running VSX and VSH are provided in the supplementary information in RosettaScripts[58] XML format.

**Improvements to the forcefield and docking protocols.** We made several general enhancements to the Rosetta General Forcefield (RosettaGenFF) and docking protocols to improve performance and accuracy. New atom types and torsion types were incorporated into RosettaGenFF to enable better modeling of three and four-membered rings. We also fixed a few issues with the torsional potential that arose from incorrect definitions of some torsion types. In addition, we corrected an error that occurred during the optimization of the tautomer state of histidine in ligand docking. Furthermore, we optimized the small molecule preprocessing scripts to address known issues and enhance their robustness. This included accurately handling the atom typing of aromatic ring nitrogens, protonated nitrogens, and oxime oxygen atoms, as well as dealing with molecules with collinear structures and others.

## Score function and virtual screening benchmarks

**CASF2016 benchmarks.** Protein structures from CASF2016 were preprocessed by relaxing the entire structure with coordinate constraints using the Rosetta FastRelax protocol[59–61]. The ligands in the CASF2016 dataset were processed using AmberTools 23.0's antechamber[62] to generate Mol2 files with am1bcc partial charges. These Mol2 files were then converted into Rosetta params files using the mol2genparams.py script in Rosetta. Since there was no sampling process for this dataset, we used the RosettaVS evaluation mode to assess the score function performance of RosettaGenFF-VS. For the scoring power test, ligands and pocket side chains were minimized with coordinate constraints. The binding affinities reported in Supplementary Fig. 2 were estimated using the locally optimized structure with two different entropy models, namely "Simple entropy" and "Default entropy". The results showed that the RosettaGenFF-VS with the "Simple entropy" model was the leading physics-based score function for binding affinity prediction. It's worth noting that $\Delta VinaRF_{20}$ used machine learning descriptors. For the docking power test, small molecule decoys were minimized within the pocket using coordinate constraints. The scores from the locally optimized structure were then used to calculate docking success. We reported the performance in Fig. 1d and Supplementary Figs. 3, 7 without specifying the entropy model since the docking power of RosettaGenFF-VS doesn't depend on the choice of the entropy model. As shown in Fig. 1d and Supplementary Fig. 3, RosettaGenFF-VS achieved leading performance in ligand docking accuracy. We then further examined binding funnels following the analysis in ref. 28. The purpose of the binding funnel analysis was to demonstrate the quality of the funnel-like shape that forms around the lowest energy minimum. Unlike docking accuracy, binding funnel analysis measures the efficiency of the energy potential in driving the conformational sampling toward the lowest energy minimum. As depicted in Supplementary Fig. 7, RosettaGenFF exhibited superior binding funnels across a broad range of ligand RMSDs, suggesting a more efficient search for the lowest energy minimum compared to other methods. For the screening power test, decoys were allowed to minimize freely within the pocket, mirroring real-world virtual screening settings. The predicted binding affinities from this process were used to calculate both the enrichment factor and success rate. The results (Fig. 1e, f and Supplementary Figs. 4, 5) showed that RosettaGenFF-VS achieved state-of-the-art performance on the screening power test. To further examine the improved performance on the screening power test, we analyzed the binding affinity prediction models with different entropy estimations on three subsets based on the excluded volume inside the binding pocket upon binding ($\Delta VOL$), buried percentage of the solvent-accessible area of the ligand

 

upon binding (ΔSAS), and hydrophobic scale of the binding pocket (H-scale). The result in Supplementary Fig. 8 indicated that both the binding affinity methods performed equally well or better on almost every subset compared to other methods. Furthermore, our method showed notable improvements on target proteins with more polar (subset H1), shallower (subset S1), and smaller (subset V1) pockets where other methods generally underperformed. In Fig. 1e and f, we reported the best performance from RosettaGenFF-VS-Simple as RosettaGenFF-VS for simplicity. Although our primary comparison is with other well-established physics-based scoring functions in this work, to provide a more comprehensive evaluation of the model's performance, we have compared our model to several state-of-the-art deep-learning based scoring functions (see Supplementary Data 1), where RosettaGenFF-VS fares favorably.

**DUD benchmarks.** The Directory of Useful Decoys (DUD) dataset was downloaded from https://dud.docking.org/https://dud.docking.org/. Receptor structures were prepared by replacing any non-canonical amino acid in the provided protein pdb file with its corresponding canonical form, followed by a coordinate constraint relax using the Rosetta FastRelax protocol. The input complex structures for RosettaVS were prepared by randomly placing a ligand molecule inside the binding pocket in each receptor to indicate the position of the binding pocket. Small molecule params files were converted from mol2 files provided by DUD. For Example, the Rosetta scripts XML file and Rosetta command line can be found in the supplemental information. The results (Fig. 1b,c) showed that both VSX and VSH modes in RosettaVS achieved superior performance compared to other virtual screening methods. On average, the VSH mode showed a slight improvement in performance compared to the VSX mode. We examined the cases where the VSH mode improved performance and presented two examples in Supplementary Fig. 10, highlighting the importance of modeling flexible sidechains in ligand docking.

## AI accelerated virtual screening protocol

**Receptor preprocessing.** For KLHDC2 virtual screening, the crystal structure of KLHDC2 in complex with SelK degron peptide (PDB: 6DO3) was downloaded from the Protein Data Bank[63]. We removed all solvent molecules and retained one copy of the degron-bound monomer structure (chains A and B). This monomer complex was relaxed using the Rosetta FastRelax protocol with coordinate constraint. As a final step, we replaced the C-end degron peptide in the relaxed structure with a random small molecule as an indication of the binding site. This modified structure was then used as the input to RosettaVS. For Nav1.7 virtual screening, the cryoEM structure of human $Na_V1.7$ - NavAb channel chimera (PDB: 5EK0)[26] was downloaded from the Protein Data Bank. After removing all solvent molecules, we retained a region of the human $Na_V1.7$ - NavAb channel chimera VSD4 (from M1493 to P1617) that directly interacts with the ligand GX-936 and used it as the receptor structure. This receptor structure was relaxed with the ligand GX-936 using the Rosetta FastRelax protocol with coordinate constraints. This resulting relaxed structure was then used as input to RosettaVS. It's important to note that GX-936 was only used to indicate the binding site for RosettaVS.

**Small molecule preprocessing and library preparation.** RosettaVS ligand docking requires Rosetta params files as input for small molecules and small molecule mol2 format is required for the generation of the params files. The Enamine REAL 2022-q1 library, which contains ~ 5.5 billion SMILES, was downloaded from Enamine (https://enamine.net/library-synthesis/real-compounds). Using dimorphite-dl[64], SMILES strings were assigned a protonation state at PH = 7.4. These properly protonated smile strings were then converted to mol2 files with molecular 3D structure and MMFF94 partial charges using OpenBabel[65]. The ZINC22 library, which contains approximately 4.1

billion ready-to-dock small molecule mol2 files, was downloaded from CartBlanche22 web server (https://cartblanche.docking.org). The *mol2genparams.py* script in Rosetta was used to convert the mol2 files into Rosetta params files for use as input in RosettaVS. To enhance the efficiency of inference on the entire library, we pre-generated fingerprints for the entire collection of the molecules, preparing them for input to the deep learning models. Utilizing RDKit[66], we generated 1024-bit Morgan fingerprints[67] with a radius of 2.

**Active learning model.** We implemented an active learning model using a simple fingerprint-based feed-forward neural network (FFN) for training target-specific classification models of binders versus non-binders. The choice of the FFN was made to reduce the cost of the surrogate model, and the model architecture and hyperparameter search are subject to future improvements. The model takes a 1024-bit fingerprint vector as input and outputs a single value representing the probability of a molecule being a binder. The FFN model consists of two densely connected hidden layers, each containing 3000 nodes. These layers are followed by batch normalization and a dropout rate of 0.5 to prevent overfitting. The final layer is linear and is followed by a sigmoid activation layer, which compresses the output values between 0 and 1, thereby representing the probability of a molecule being a binder. With precomputed fingerprints, the inference of one million molecules using this model will take, on average, around 110 s using a single CPU (Intel Xeon E5-2695 v3 @ 2.30 GHz), or around 11 s on average using an RTX2080 GPU. The model performance was monitored throughout the active learning process, and the final model for KLHDC2 has an AUC of 0.886, and the final model for the Nav1.7 target has an AUC of 0.927 on an independent test set, which shows the final ML model is indeed a good binary classifier for each target.

**OpenVS platform.** We developed an open-source platform, OpenVS, to streamline the entire AI-accelerated virtual screening process. A key feature of this platform was its ability to enable the parallelization of the virtual screen and reduce the load on the file system. This was achieved by batching multiple ligands into a single virtual screen job through the multi-ligand docking feature in the RosettaVS protocol. By docking N ligands in a single job, the number of input and output files was effectively reduced by a factor of N, significantly decreasing the load on the file system. In addition, batching multiple ligands together also reduced the input and output (I/O) load on the system since fewer jobs were required to run in parallel. The platform used the SLURM workload manager (https://slurm.schedmd.com) and GNU parallel[68] for high parallelization of virtual screens, exhibiting linear scaling of virtual screening time relative to the number of CPUs and nodes utilized. Furthermore, this platform could be easily adapted to support other job schedulers, making it a versatile tool for various computational environments.

**Workflow of AI accelerated virtual screening.** In this work, we utilized active learning techniques to guide the exploration of the vast chemical space. The greedy strategy was used to select new compounds for each iteration to augment the training dataset without using any explicit uncertainty information to reduce the computational cost and inference time. In principle, uncertainty estimation can be obtained using Monte Carlo (MC) dropout[69] by running model inference multiple times with activated dropout layers within the current framework. The concrete workflow is described as follows. Our first step was to create a specialized subset, referred to as the druglike-centroid library, from the ZINC15[70] 3d druglike database containing ~ 493.6 million molecules. The creation of this subset involved clustering similar molecules from the ZINC15 3D druglike database, using a cutoff of 0.6 Tanimoto similarity. From each cluster, the smallest molecule was selected and added to the library, serving as the centroid of the cluster. This process resulted in the formation of the

druglike-centroid library, which includes around 13 million molecules. The purpose of creating the druglike-centroid library was to ensure that the model was exposed to a wide range of chemical space during the initial iteration. For the first iteration, 0.5 million and 1 million molecules were randomly selected as the training and testing datasets, respectively, from the druglike-centroid library. We used the VSX mode in RosettaVS to dock these molecules to the target pocket. Morgan fingerprints using a radius of 2 and 1024 bits were generated for all the molecules as the input to the deep learning model. We selected a predicted binding affinity (dG) cutoff corresponding to the top N% of the molecules in the testing set, for each iteration. This cutoff was used to assign binders (<= dG_cutoff) or non-binders (> dG_cutoff) labels to the docked molecules. We used ten log spaces between 10% to 0.01% to set the top N% for each iteration. For example, the first iteration had N% = 10%, the second iteration had N% = 4.64%, and the tenth iteration had N% = 0.01%. The model was trained as a target-specific binary classifier to predict whether a molecule is a binder or not. We computed the cross-entropy loss on the testing set for each epoch and used early stopping to prevent the model from overfitting to the training set. We used the model to predict the entire Enamine REAL 2022-q1 library (~ 5.5 billion compounds) or ZINC22 library (~ 4.1 billion compounds). From these predictions, we selected the top-ranked 0.25 million and an additional 0.25 million randomly selected molecules for the next iteration docking. In the next iteration, the newly selected 0.5 million molecules were docked to the pocket using VSX mode. These 0.5 million molecules, combined with previous molecules in the training set, were used to train a new model. A tighter dG cutoff was selected corresponding to the current iteration's top N% in the testing dataset. After training a new model, we used it to predict the entire library again and selected another 0.5 million molecules for the next iteration. We repeated this process until the maximum number of iterations (ten iterations) was reached or the predicted binding affinities of top-ranked molecules converged. The top-ranked 50,000 or 100,000 molecules from the virtual screening were re-docked using VSH mode to account for the flexibility of the receptor. (See Supplementary Fig. 1 for the flowchart of this protocol.)

**Filters for selecting promising compounds.** We computed the log octanol/water value of the partition coefficient of the compound (cLogP), the number of unsatisfied hydrogen bonds (Nunsats) at the interface between the protein and ligand, and the number of torsion angle outliers (N_unusual_torsion) from CSD torsion geometry analysis using RDKit, Rosetta[71], and CSD[72] python package respectively. For the KLHDC2 virtual screen, docked poses with cLogP > 3.5 Nunsats > 1 were discarded. Similarly, for the Na$_V$1.7 virtual screen, docked poses with cLogP > 3.5, Nunsats > 1, and N_unusual_torsion > 1 were discarded. These filters helped us in reducing the false positives and refine our selection of compounds for final experimental validation.

**Filtering and clustering.** To reduce the redundancy of the molecules selected for experimental validation, we clustered the top-ranking molecules from both screens based on Tanimoto similarity. For the KLHDC2 virtual screen, we took the top-ranked 1000 compounds and filtered out compounds with low predicted solubility (removing 93 compounds) and unsatisfied hydrogen bonds in the bound conformation (removing 754 compounds), yielding 153 compounds. We then applied clustering with a Tanimoto similarity cutoff of 0.6 to remove similar molecules, which resulted in 54 clusters, each represented by the member with the best binding affinity. These 54 cluster representative molecules were then subjected to manual inspections. For the Na$_V$1.7 screen, we clustered the top 100,000 molecules with a cutoff of 0.6 Tanimoto similarity. This resulted in 16820 clusters, each represented by the member with the best predicted binding affinity. The top 1000 ranked cluster representative molecules were subjected to the filtering process where we removed molecules with low

predicted solubility (removing 183 compounds), unsatisfied hydrogen bonds in the bound conformation (removing 139 compounds), and torsion angle outliers (removing 520 compounds) in Cambridge Structural Database (CSD)[72]. A total of 160 molecules passed the filtering were examined manually. The filters for unsatisfied hydrogen bonds and torsion angle outliers are the two that removed the most molecules. Although the current energy model achieves state-of-the-art performance, our filters suggest that these conformations are 'under penalized' by the current energy model, indicating an opportunity for improvement in the scoring function.

**Rescreening against KLHDC2.** To further validate the reliability of RosettaVS for hit discovery, we performed a rescreening experiment against the KLHDC2 target using the Enamine Screening library. This library contains around 4.1 million compounds, which are the first samples of billions of synthesizable compounds. Our confirmed best-hit compound, C2.8, was included in this screening library. We were able to rediscover compound C2.8 from this experiment. In fact, it was among the top 1000 compounds, which is within 0.024% of the 4.1 million compounds, before any filtering. After applying the buried unsatisfied hydrogen bond filter, we removed any docked complex that had more than one buried polar atom. Following this, the hit compound C2.8 was among the top 50 compounds. The success of this rescreening experiment shows our virtual screening protocol is both effective and reliable for hit discovery.

**Common chemical properties.** We have reported several chemical properties that are important in small molecule drug discovery, including the quantitative estimate of drug-likeness (QED)[73], the calculated octanol-water partition coefficient (cLogP)[74], and synthetic accessibility (SA)[75]. As shown in Supplementary Table 2, the compounds ordered for both targets have an average QED above 0.7, indicating a high degree of drug-likeness. The cLogP values of the compounds fall within a reasonable range, as it is used as a filter to remove compounds that are excessively hydrophobic (see 'Filters for selecting promising compounds'). The average SA is low, as expected because all the compounds from Enamine REAL or ZINC22 are highly synthesizable.

## Experimental methods for KLHDC2

**Molecular biology and protein purification.** For DNA extraction, E.coli DH5α was grown for 16 hr at 37 °C. For bacmid production, E.coli DH10Bac was grown for 16 hr at 37 °C. For baculovirus production and amplification, Sf9 (LifeTechnologies, B82501) insect cells were grown for 2–3 days at 26 °C. For protein expression, both E.coli BL21(DE3) (grown for 16 hr at 18 °C) and HighFive insect cells (grown for 2–3 at 26 °C, 105 RPM) were used. LB Broth Miller (Fisher BioReagents) was used for E.coli. Sf9 insect cells were maintained in Grace's Insect Medium (Gibco) supplemented with 7% FBS (Gibco) and 1% Penicillin-Streptomycin (HyClone) solution. Suspension HighFive(LifeTechnologies, B85502) cells were grown in EXPRESSTM FIVE SFM(Gibco) supplemented with 5% L-Glutamine 200 mM (HyClone) and 1% Penicillin-Streptomycin (HyClone) solution. Tissue culture media and supplements were from GIBCO Life Technologies (Carlsbad, CA, USA). Cells have been authenticated by the vendors. No further authentication was performed for commercially available cell lines.

The kelch repeat domain of human KLHDC2 (UniProt: Q9Y2U9, amino acid 1–362) was subcloned into the pFastBac vector with an N-terminally fused glutathione-S-transferase (GST), and a TEV-cleavage site. A recombinant baculovirus was produced and amplified three times in Sf9 monolayer cells to produce P4. The P4 virus was used to infect HighFive suspension insect cell cultures to produce the recombinant GST-KLHDC2 protein. The cells were harvested 2–3 days post-infection, re-suspended, and lysed in lysis buffer (20 mM Tris, pH

8.0, 200 mM NaCl, 5 mM DTT) in the presence of protease inhibitors (1 µg/ml Leupeptin, 1 µg/ml Pepstatin and 100 µM PMSF) using a microfluidizer. The GST-KLHDC2 protein was isolated from the soluble cell lysate by PierceTM Glutathione Agarose (Thermo Scientific). For AlphaLISA competition assays, the GST-tagged KLHDC2 was further purified by Q Sepharose High-Performance resin (GE Healthcare). The NaCl eluates were subjected to a Superdex-200 size exclusion chromatography column (GE Healthcare). All samples were flash-frozen in liquid nitrogen for storage prior to use. For protein crystallization, the kelch repeat domain of human KLHDC2 (amino acids 22–362) was subcloned into the pET vector with an N-terminally fused His-elongation factor Ts (TSF) and a TEV-cleavage site. The His-TSF-KLHDC2 protein was overexpressed and purified from BL21 (DE3) E. coli cells. Bacterial cells transformed with the pET-based expression plasmid were grown in LB broth to an OD600 of 0.8–1 and induced with 0.5 mM IPTG. Cells were harvested, re-suspended, and lysed in lysis buffer (20 mM Tris, pH 8.0, 200 mM NaCl, 20 mM imidazole) in the presence of protease inhibitors (1 µg mL−1 leupeptin, 1 µg mL−1 pepstatin and 100 µM phenylmethylsulfonyl fluoride) using a microfluidizer. The His-TSF-KLHDC2 protein was isolated from the soluble cell lysate by HisPurTM Ni-NTA Superflow Agarose (Thermo Fisher Scientific, Waltham, Massachusetts). After TEV cleavage of the His-TSF, KLHDC2 was further purified by Mono Q™ 5/50 GL (GE Healthcare, Chicago, Illinois). The NaCl eluates were subjected to Superdex-200 size-exclusion chromatography (GE Healthcare). Native mass spectrometry was used to confirm KLHDC2 in its apo form. The samples were concentrated by ultrafiltration to 10–22 mg mL−1. All samples were flash-frozen in liquid nitrogen for storage prior to use.

**Protein crystallization.** The crystals of KLHDC2 in its apo form were grown at 25 °C by the hanging-drop vapor diffusion method with 2 parts protein sample to 1 part reservoir solution containing 0.03 M MgCl$_2$*6H$_2$O, 0.03 M CaCl$_2$*2H$_2$O, 10% (w/v) PEG 20000, 20% (v/v) PEG MME, 0.1 M Tris (base)/ bicine pH 8.5. Crystals of maximal sizes were obtained and harvested after a few days. Cryoprotection was provided by the crystallization condition.

**Data collection and structure determination.** After collecting native datasets at Advanced Light Source Beamline 8.2.1 (Data collection: exposure – 0.4 sec, energy – 12397.1 keV, wavelength – 1 Å, temperature – 100 K), X-ray diffraction data was automatically processed using xia2 to run DIALS 3.8. The structures were solved by molecular replacement using the kelch domain of KLHDC2 (PDB:6DO3) with Phaser from the PHENIX suite of programs package version 1.20.1. All structural models were manually built, refined, and rebuilt with COOT[76] version 0.9.8.91 and PHENIX[77] version 1.20.1. PyMOL[34] version 2.5.5, and LIGPLOT[78] version 2.0 were used to generate figures. After refinement, the Ramachandran statistics are as follows : Ramachandran favored - 97.62 %, Ramachandran allowed - 2.38 %, Ramachandran outliers - 0.00 %, Rotamer outliers - 0.36 %.

**AlphaLISA luminescence proximity assay.** AlphaLISA assays for determining and measuring protein-protein interactions were performed using an EnSpire reader (PerkinElmer). GST-tagged KLHDC2 was attached to anti-GST AlphaLISA acceptor beads. Synthetic N-terminal biotinylated 12 aa SelK degron peptide ([Biotin] HLRGSPPPMAGG[C] (Bio-Synthesis, Inc.) was immobilized to streptavidin-coated AlphaLISA donor beads. The donor and acceptor beads were brought into proximity by the interactions between the SelK peptide and KLHDC2. Excitation of the donor beads by a laser beam of 680 nm promotes the formation of singlet oxygen. When an acceptor bead is in close proximity, the singlet oxygen reacts with thioxene derivatives in the acceptor beads and causes the emission of 615 nm photons, which are detected as the binding signal. If the beads are not in close proximity to each other, the oxygen will return to its

ground state, and the acceptor beads will not emit light. Competition assays were performed in the presence of numerous compounds, which were titrated at various concentrations. The experiments were conducted with 3.83 nM of GST-KLHDC2 and 5.55 nM biotinylated 12 aa SelK peptide in the presence of 5 µg/ml donor and acceptor beads in a buffer of 25 mM HEPES, pH 7.5, 100 mM NaCl, 1 mM TCEP, 0.1% Tween-20, and 0.05 mg/ml Bovine Serum Albumin. The compound concentrations used in competition assays ranged from 15 nM to 1.5 mM. The experiments were done in triplicates. IC$_{50}$ values were determined using non-linear curve fitting of the dose-response curves generated with Prism 8 (GraphPad).

**Octet Bio-Layer Interferometry measurement.** Octet BioLayer interferometry competition assay monitoring the ability of C29 to interfere with the binding between the biotinylated 12 aa SelK peptide and GST-KLHDC2 was monitored using the Octet Red 96 (ForteBio, Pall Life Sciences) following the manufacturer's procedures. The reaction was carried out in black 96 well plates maintained at 30 °C. The reaction volume was 200 µL in each well. The Octet buffer used throughout the experiment contained 20 mM Tris-HCl, 200 mM NaCl, 5 mM DTT and 0.1% BSA, pH 8.0. The Loading buffer contained the Octet buffer and biotinylated 12 aa SelK peptide, at a final concentration of 200 nM. The Quench buffer contained the Octet buffer and biocytin, at a final concentration of 0.1 mM. The Association buffer contained the Octet buffer, GST-KLHDC2, at a final concentration of 50 nM, and the competitor small molecule - C29 at various final concentrations (30 µM, 10 µM, 3.33 µM, 1.11 µM or 0 µM). Prior to binding analysis, the streptavidin coated optical probes were incubated in Octet buffer for 60 sec, loaded for 96 sec with biotinylated 12 aa SelK peptide (Loading buffer), quenched for 60 s with biocytin (Quench buffer) and baselined in Octet buffer for 60 sec. The binding of the analyte GST-KLHDC2 in the presence or absence of the C29, to the optical probes was measured simultaneously using instrumental defaults for 225 sec as the probes were then incubated in Association buffer. The dissociation was measured for 600 sec while the probes were incubated in Octet buffer. While not loaded with ligand, the control probes were quenched. There was no binding of analyte and competitor to the unloaded probes. The data were analyzed by the Octet data analysis software version 9.0.. The association and dissociation curves were fitted with a Local Full fit, 1:1 ligand model. The data was plotted using Excel. The IC$_{50}$ value was calculated using Prism 8 (GraphPad) from the nonlinear regression curve fit of the response values vs. log competitor concentration.

**Experimental methods for Na$_V$1.7**
**Cell cultures and electrophysiology.** HEK-293 cells stably expressing human Na$_V$1.5 and Na$_V$1.7 were obtained from Dr. Chris Lossin (UC Davis, CA). HEK293 cells stably expressing HERG (hKv11.1) were a gift from Craig January (University of Wisconsin, Madison). Cells were cultured in complete DMEM supplemented with 10% FBS, 1% penicillin/ streptomycin, and G418. All manual whole-cell patch-clamp recordings were performed at room temperature (22–24 °C) using an EPC-10 amplifier (HEKA Electronik, Lambrecht/Pfalz, Germany) on cells that were grown to 60–80% confluency, lifted with TrypLE, and plated onto poly-l-lysine–coated coverslips. External bath solution contains (in mM) 160 NaCl, 4.5 KCl, 2 CaCl$_2$, 1 MgCl$_2$, 10 HEPES (pH 7.4 and 305 mOsm) as. Patch pipettes were heat-pulled from soda lime glass (micro-hematocrit tubes, Kimble Chase, Rochester, NY) and had resistances of 2–3 MΩ when filled with an internal solution. Patch recordings of Na$_V$1.5 and Na$_V$1.7 channels were done using a cesium fluoride-based internal solution containing (in mM) 10 NaF, 110 CsF, 20 CsCl, 10 HEPES, 2 EGTA, (pH 7.4, 310 mOsm). Recordings of HERG channels were conducted using a potassium fluoride-based internal solution with added ATP (160 mM KF, 2 mM MgCl$_2$, 10 mM EGTA, 10 mM HEPES, 4 mM NaATP, pH = 7.2 and 300–320 mOsm). Data

acquisition and analysis were performed with Pulse-PulseFit (HEKA Electronik GmbH, Germany), IgorPro (WaveMetrics, Portland, OR), and Origin 9.0 software (OriginLab Corporation, Northampton, MA). To investigate specifically the inactivated state block of $Na_V1.7$ and $Na_V1.5$, holding potentials were set at the $V_{1/2}$ of maximal inactivation ($-70$ mV and $-80$ mV, respectively) before voltage stepping to $-150$ mV for 20 msec, and then 0 mV for 50 msec to elicit inward currents. Inhibition of currents in the resting state was recorded using a holding potential of $-120$ mV. Control test currents were monitored for up to 5–10 min to ensure that the amplitude and kinetics of the response were stable. Series resistance was compensated to 80–90% and linear leak currents and capacitance artifacts were corrected using a P/4 subtraction method. The pulse interval was 0.1 Hz. For all experiments, stock solutions of the drugs were prepared fresh from 10 mM stocks in DMSO and test solutions prepared in external bath solutions were applied to individual cells into the recording bath. For measuring inhibition, currents were allowed to saturate with repeated pulsing before addition of subsequent doses. IC50 values were derived from measurements performed on individual cells that were tested with at least three or more concentrations of each peptide. For the HEK-239 HERG cells, a 2-step pulse (applied every 10 sec) from $-80$ mV first to 40 mV for 2 sec and then to $-60$ mV for 4 sec, was used to elicit HERG currents. The percent reduction of HERG tail current amplitude by the compounds was determined. The percent of a block of tail current amplitude by the drugs was determined and data are shown as mean +/$-$ SD ($n$ = 3–4 per data point).

**Compound synthesis.** Chemical synthesis for all compounds in this work was performed by Enamine within 5 weeks with over 90% purity and used without further purification. Purity of compounds were assessed based on liquid chromatography–mass spectrometry (LC-MS). See Supplementary Data 2-3 for purity and LC-MS spectra.

### Reporting summary
Further information on research design is available in the Nature Portfolio Reporting Summary linked to this article.

## Data availability
All compounds in this work are available from Enamine. The virtual screening library, Enamine REAL, and ZINC22, used in this work are freely available from Enamine [https://enamine.net/] or ZINC22 [https://cartblanche.docking.org/], respectively. The druglike-centroid library generated in this study is available for download at https://files.ipd.uw.edu/pub/OpenVS/centroids.tgz [https://files.ipd.uw.edu/pub/OpenVS/centroids.tgz]. The source data for figures and tables in the main text and supplementary information are provided in the Supplementary Information/Source Data file. The crystal structure of the KLHDC2-C29 complex is available from the PDB with accession code: 8UXS. Chemical purity of all the compounds and their LC-MS spectra are available in Supplementary Data 2-3. Source data are provided in this paper.

## Code availability
RosettaVS is within Rosetta software, which is freely available for non-commercial research at https://github.com/RosettaCommons/rosetta. OpenVS code and scripts are available on Zenodo[79] and also on GitHub at https://github.com/gfzhou/OpenVS.

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

## Acknowledgements

F.D. and G.Z. are supported by funds from the DARPA Harnessing Enzymatic Activity for Lifesaving Remedies (HEALR) program (HR001120S0052 contract HR0011-21-2-0012) and the Defense Threat Reduction Agency (DTRA) (HDTRA1-22-1-0012). D.C. and M.F.B. are supported by the National Science Foundation through awards 1807382 and 2203513 from the Division of Chemistry with partial co-funding from the Division of Molecular and Cellular Biosciences. H.P. is supported by National Research Foundation of Korea (NRF) grants (No. 2022R1C1C1007817). L.S. is supported by the Washington State funding. D.V.R. and N.Z. are supported by Howard Hughes Medical Institute. N.Z. is a Howard Hughes Medical Institute Investigator. We thank the beamline staff of the Advanced Light Source (ALS) at the University of California at Berkeley and the Advanced Photon Source (APS) at Argonne National Laboratory for help with data collection. Beamlines 8.2.1 and 8.2.2 of the Advanced Light Source, a DOE Office of Science User Facility under Contract No. DE-AC02-05CH11231, is supported in part by the ALS-ENABLE program funded by the National Institutes of Health, National Institute of General Medical Sciences, grant P30 GM124169-01. This research used resources of the Advanced Photon Source, a U.S. Department of Energy (DOE) Office of Science user facility operated for the DOE Office of Science by Argonne National Laboratory under Contract No. DE-AC02-06CH11357.

## Author contributions

G.Z. and F.D. designed the study. G.Z., H.P., and F.D. developed the computational methods. G.Z. conducted the benchmark experiments. D-V.R., N.Z., H.M.N., H.W., and V.Y-Y. provided experimental binding data. D-V.R. and N.Z collected crystal data. G.Z., D-V.R., N.Z., and F.D. drafted the original manuscript. G.Z., D-V.R., H.P., D.C., H.M.N., L.S., M.F.B., P.T.N., H.W., V.Y-Y., N.Z., and F.D. edited and reviewed the manuscript before submission.

## Competing interests

N.Z. is a scientific cofounder of and has financial interests in SEED Therapeutics. N.Z. serves as a member of the scientific advisory board of Synthex with financial interests and has received research funding from and is a shareholder of Kymera Therapeutics. The findings presented in this manuscript were not discussed with any person in these companies. The authors declare no other competing interests. All other authors declare no competing interests.
