## [Peer Review File · Nature Communications]

An artificial intelligence accelerated virtual screening platform for drug discoveryREVIEWER COMMENTS

Reviewer #1 (Remarks to the Author):

Zhou et al describe an improved method for discovering novel small molecules for challenging drug targets by virtual methods. The work has clearly been conducted to a very high standard and I have no doubt that these results will be of considerable interest to the drug discovery community at large. As such, I would certainly recommend this for publication in *Nature Communications*.

Of note, the authors are correct to point out that many similar services reside behind paywalls (such as CCDC GOLD – a situation I find particularly irksome in light of the fact that the CCDC database is entirely the result of data freely gifted by the scientific community), and thus the open-source platform reported herein is to be commended. Furthermore, the authors report the first known small molecule binders of a hitherto un-liganded E3 ligase, a discovery which will surely be of significant interest to the PROTAC and molecular degrader communities, as well as those focussed on E3 ligase biology specifically. Of note is that E3 ligases are known to be challenging to target with small molecules, likely in part at least due to the often shallow and solve-exposed binding sites that they offer. The 2nd target included here, an ion channel, also belongs to a notoriously challenging protein class, both to study (the biology of) and to target with small molecules.

I have a few minor comments which will follow below. My only perhaps non-minor comment is whether a 3rd exemplar protein class as a target could, or should, be included here. I'm thinking particularly whether the method described here would attract more attention from the drug discovery community if a more classical enzyme drug target (such as a kinase) was included in the target list (I appreciate that KLHDC2 is an E3 ligase and therefore is an enzyme but the kelch-like domain is really only involved substrate recognition and not catalysis). Could the authors please comment on this?

Some minor comments/suggestions:

1) Line 246-7: this statement will have to be modified I'm afraid, or, some evidence provided that progress (with respect to activity improvement) can be made for the C29 series. This could perhaps be achieved through SAR-by-catalogue, eg. I'm contesting this statement as E3 ligases really have been very challenging to target with small molecules, and even for the notorious players – eg VHL and cereblon – the compounds which bind to these ligases are only really sub- μ M, and not convincingly nM. So, optimising the C29 series into nM binders is not a given. (Though they are remarkable, of course, in having been discovered by the methods described herein.)

2) Re. the xray data – to be clear, the provision of a validating xray structure is commendable. A small comment, but in the legend of Figure 2, could the map type and sigma level at which it's displayed please be provided?

3) A 2nd comment on the xray data (and not a request) – in your "Table 1", your I/sigI is very high for the high res shell, and I suspect there is useful data to a higher res, particularly if a

direct-photon measuring detector was used (which I guess is likely). Again, this isn't a request for the structure to be re-refined, but if there was appetite to do this, I suspect modern opinion would allow a higher res dataset to be produced from this data (using eg $cc1/2$ to define resolution cutoff).

I have one final comment, and this is really to satisfy a curiosity I have – it is mentioned that the ability to account for protein mobility during the docking stages is likely to contribute to the improved performance of RosettaGenFF-VS over its competitors – is the mobility that can be accounted for restricted to that of side-chains, or can larger movements (such as that of backbones) be tolerated/accommodated? If no, are there plans to work towards this in future iterations of this screening platform?

Finally: extremely minor, but since I noticed it, line 244, I think "Remarkable" should be "Remarkably" perhaps.

Reviewer #2 (Remarks to the Author):

The authors present a high throughput virtual screening platform for drug discovery, which relies on i) their modifications to the physics-based Rosetta general force field and ii) an active learning driven search through chemical space.

The accuracy of the docking methods are carefully benchmarked using CASF datasets and shown to out-perform other commonly used physics-based scoring functions. It might be argued that they should also benchmark against the latest deep learning models too, but I tend to agree with their argument that it is unknown how well these model generalize outside the benchmark sets.

The workflow identifies novel low micromolar hit compounds against two targets, with binding confirmed with crystallography in one case. Although the computational resource required to achieve this is high (3000 cpus for a week), the methods are generalizable, show strong promise and should be of wide interest to the computational and drug discovery communities. I recommend publication subject to addressing the following points:

-p6. Particularly for the second case, the predicted binding free energies of around -18 kcal/mol are highly over-estimated compared to the experimental binding affinity (around micromolar). The authors should clarify if this energy scale is arbitrary or has been trained to match experimental free energy estimates.

- p8. Please give full details of how the weights on L350 are trained. What is the objective from the DUD-E set that was optimized? What force field is used in the Monte Carlo simulations?

- p9, L392. Presumably $DG = DH - TDS$ was used? If so, what temperature is used?

- p9. A few more details are needed around which terms are kept and which are removed

from RosettaGenFF. Is the binding energy a sum of Coulomb + LJ terms? No mention is made of the solvation term, so has this been removed? If that is the case, what desolvation penalty is present in the scoring function? I would expect optimization in the absence of this term to produce highly polar ligands that would also be highly water soluble.

- p14. Maybe related to the above, is it problematic that only 160 out of 1000 compound pass simple filters? This seems to indicate the post-processing is just as important a factor in compound design as the scoring function. Which criterion removed the most molecules?

- p12. Deployment of the active learning model on the entire enamine library is an impressive calculation. Can the authors please comment on the computational resources and cost of this step in particular? What is the accuracy of the final ML model at the task of predicting docking scores?

Reviewer #2 (Remarks on code availability):

I have not installed and run the code. However I have reviewed the supplied data, README files and installation instructions. Everything seems to be thoroughly documented and complete, I just have one comment:

- p10. Please clarify whether the new force field parameters are publicly available.

Reviewer #3 (Remarks to the Author):

The manuscript titled "An artificial intelligence accelerated virtual screening platform for drug discovery" by Guangfeng Zhou and colleagues introduces a cutting-edge structure-based virtual screening method known as RosettaVS. This study is poised to enhance the efficiency and precision of early-phase drug discovery efforts by accurately predicting the binding poses and affinities of compounds from extensive chemical libraries. The authors have developed an open-source platform, OpenVS, which harnesses the power of artificial intelligence to expedite the virtual screening process. The RosettaVS method shows promise in surpassing current state-of-the-art techniques, with the capacity to manage libraries comprising billions of compounds. The platform's efficacy was demonstrated through its application to two targets: KLHDC2, a novel ubiquitin ligase, and the human voltage-gated sodium channel NaV1.7, yielding several novel hits with single-digit micromolar binding affinities within a remarkably short timeframe of one week.

Despite the innovative nature of the approach to virtual screening and drug discovery, the manuscript raises several concerns regarding the model's innovativeness, precision, and presentation of results. The manuscript does not convincingly establish the novelty and accuracy of the RosettaVS model when benchmarked against existing, well-established methods. To fortify the study's claims, we suggest a comprehensive comparison with current leading-edge techniques, which would yield a more objective evaluation of the model's performance.

The following specific areas require further attention:

The manuscript would be strengthened by the incorporation of permutation tests or other robust statistical methodologies to evaluate the significance of the model's performance. Such methodologies would lend rigor to the interpretation of the findings and bolster the model's claimed effectiveness.

The manuscript should provide a more detailed explanation of the uncertainty estimation methods employed in the active learning component of the OpenVS platform. Transparency regarding these methods is essential for grasping the dependability of the model's predictions and ensuring the reproducibility of the research.

The study could be markedly improved by integrating key drug-likeness metrics, such as logP, QED score, or Synthetic Accessibility score, into the model. These parameters are pivotal for assessing drug-likeness and should be thoroughly examined in the context of the KLHDC2 and Nav1.7 case studies for a holistic evaluation.

We strongly advocate for the execution of redocking experiments for the virtual screening of KLHDC2 and Nav1.7. This additional validation step is crucial for confirming the model's predictions and ensuring the reliability of the identified binders.

Lastly, the manuscript's figure presentation requires refinement. The figures must conform to the journal's guidelines for clarity, resolution, and informative labeling. It is imperative to ensure that all visual components are of the highest quality and accurately depict the data under communication.

REVIEWER COMMENTS

Reviewer #1 (Remarks to the Author):

Zhou et al describe an improved method for discovering novel small molecules for challenging drug targets by virtual methods. The work has clearly been conducted to a very high standard and I have no doubt that these results will be of considerable interest to the drug discovery community at large. As such, I would certainly recommend this for publication in nature communications.

Of note, the authors are correct to point out that many similar services reside behind paywalls (such as CCDC GOLD – a situation I find particularly irksome in light of the fact that the CCDC database is entirely the result of data freely gifted by the scientific community), and thus the open-source platform reported herein is to be commended. Furthermore, the authors report the first known small molecule binders of a hitherto un-liganded E3 ligase, a discovery which will surely be of significant interest to the PROTAC and molecular degrader communities, as well as those focussed on E3 ligase biology specifically. Of note is that E3 ligases are known to be challenging to target with small molecules, likely in part at least due to the often shallow and solve-exposed binding sites that they offer. The 2nd target included here, an ion channel, also belongs to a notoriously challenging protein class, both to study (the biology of) and to target with small molecules.

I have a few minor comments which will follow below. My only perhaps non-minor comment is whether a 3rd exemplar protein class as a target could, or should, be included here. I'm thinking particularly whether the method described here would attract more attention from the drug discovery community if a more classical enzyme drug target (such as a kinase) was included in the target list (I appreciate that KLHDC2 is an E3 ligase and therefore is an enzyme but the kelch-like domain is really only involved substrate recognition and not catalysis). Could the authors please comment on this?

We thank the reviewer for the suggestion. For this manuscript, we intentionally chose an E3 ligase and an ion channel as two targets that seemed both challenging, and diverse, to illustrate the effectiveness of our computational design methodology. We are currently working on applying this to a number of other targets, including kinases as well as other conventional enzymes. However, we feel these efforts are beyond the scope of this manuscript; it is our plan to publish the screening results for some of these classic targets in the near future.

Some minor comments/suggestions:

1) Line 246-7: this statement will have to be modified I'm afraid, or, some evidence provided that progress (with respect to activity improvement) can be made for the C29 series. This could perhaps be achieved through SAR-by-catalogue, eg. I'm contesting this statement as E3 ligases really have been very challenging to target with small molecules, and even for the notorious players – eg VHL and cereblon – the cmpds which bind to these ligases are only really sub-uM, and not convincingly nM. So, optimising the C29 series into nM binders is not a

given. (Though they are remarkable, of course, in having been discovered by the methods described herein.)

At this moment, we actually have already obtained a C29 derivative that binds KLHDC2 with an affinity in the nM range. We are developing a functional PROTAC based on this binder and plan to publish the results in a separate manuscript. Nevertheless, we have revised the statement to be more neutral without referring to the progress we have made so far.

Here is the revised sentence in the manuscript:

“Future optimization will be needed to improve the potency of these compounds to reach the nanomolar range.”

2) Re. the xray data – to be clear, the provision of a validating xray structure is commendable. A small comment, but in the legend of Figure 2, could the map type and sigma level at which it's displayed please be provided?

We thank the reviewer for reminding us to be more precise with the figure legend. We have now indicated that an mFo-DFc map contoured at 1.5 sigma and calculated before the compound was built into the model is shown in the figure.

3) A 2nd comment on the xray data (and not a request) – in your “Table 1”, your I/sigI is very high for the high res shell, and I suspect there is useful data to a higher res, particularly if a direct-photon measuring detector was used (which i guess is likely). Again, this isn't a request for the structure to be re-refined, but if there was appetite to do this, I suspect modern opinion would allow a higher res dataset to be produced from this data (using eg cc1/2 to define resolution cutoff).

We appreciate the reviewer's insightful comment and completely agree with the reviewer that there is useful data at a higher resolution, especially if we use cc1/2 to define the resolution cutoff. In fact, this is what we are currently using during compound optimization.

I have one final comment, and this is really to satisfy a curiosity I have – it is mentioned that the ability to account for protein mobility during the docking stages is likely to contribute to the improved performance of RosettaGenFF-VS over its competitors – is the mobility that can be accounted for restricted to that of side-chains, or can larger movements (such as that of backbones) be tolerated/accommodated? If no, are there plans to work towards this in future iterations of this screening platform?

While our protocol allows full pocket sidechain flexibility, it only allows for very limited movement of the backbone, as described in Park et al (JCTC 2021). The current protocol does not support large movement of protein backbone in docking. That said, accurate modeling of large conformational changes induced upon binding is still an open and changeling problem in the field and we are working on it in our future developments. We have tried to be more clear with this in the text.

“In addition, we adapted a docking protocol from our previous work that allows for a significant degree

of receptor flexibility, allowing us to model flexible sidechains as well as limited backbone movement in our virtual screening protocol.”

Finally: extremely minor, but since I noticed it, line 244, I think “Remarkable” should be “Remarkably” perhaps.

We thank the reviewer for noticing this typo. It’s been fixed in the revision.

Reviewer #2 (Remarks to the Author):

The authors present a high throughput virtual screening platform for drug discovery, which relies on i) their modifications to the physics-based Rosetta general force field and ii) an active learning driven search through chemical space.

The accuracy of the docking methods are carefully benchmarked using CASF datasets and shown to out-perform other commonly used physics-based scoring functions. It might be argued that they should also benchmark against the latest deep learning models too, but I tend to agree with their argument that it is unknown how well these model generalize outside the benchmark sets.

The workflow identifies novel low micromolar hit compounds against two targets, with binding confirmed with crystallography in one case. Although the computational resource required to achieve this is high (3000 cpus for a week), the methods are generalizable, show strong promise and should be of wide interest to the computational and drug discovery communities. I recommend publication subject to addressing the following points:

-p6. Particularly for the second case, the predicted binding free energies of around -18 kcal/mol are highly over-estimated compared to the experimental binding affinity (around micromolar). The authors should clarify if this energy scale is arbitrary or has been trained to match experimental free energy estimates.

We thank the reviewer for pointing it out. Yes, the absolute energy scale is arbitrary. We have revised the “Binding affinity estimation” section in the manuscript to make it more clear to the readers :

“It’s worth noting that the scale of the estimated binding affinity is arbitrary, and it doesn’t directly correspond to the experimental binding affinity.”

- p8. Please give full details of how the weights on L350 are trained. What is the objective from the DUD-E set that was optimized? What force field is used in the Monte Carlo simulations?

We apologize for not being clear on this. The force field used in the MC simulations is RosettaGenFF.

We have revised the manuscript to include full details of how the optimal weights were obtained.

“To estimate the probability distribution of each rotatable torsion value at unbound state, conformations from every 3,000 steps of ligand-only Monte Carlo (MC) simulation using RosettaGenFF at room temperature (300K) are collected.”

“The optimal weights for ΔS_{rb} and $\Delta S_{torsion}$ were obtained using a grid search to maximize the AUROC of a non-overlapping subset of DUD-E set⁵⁴ (see SI, ‘Subset of DUD-E’ for more details). The weights (w_1, w_2) considered in this search were $\{(0,0), (0,1), (1,0), (1,1), (0,2), (2,0), (2,1), (1,2), (2,2)\}$ ”

In the Supplementary Methods, we added:

“**The subset of DUD-E.** The subset of DUD-E used for determining the optimal weights for the default entropy model contains ten targets that are randomly selected from the targets with around 200 actives. And these targets are *abl1, aofb, cp2c9, def, fpps, hivint, kit, mcr, thb, xiap.*”

- p9, L392. Presumably $\Delta G = \Delta H - T\Delta S$ was used? If so, what temperature is used?

The explicit temperature used was room temperature (300K) in estimating ΔS by running MC simulation and converting the ligand torsion distribution to the torsional entropy:

$$\Delta S_{torsion} = RT \sum_i \sum_j (-p_{ij} \log p_{ij})$$

The temperature in the ΔG estimation is implicitly included in the ΔS estimation. We have revised the manuscript to be more clear on this:

“Monte Carlo (MC) simulation using RosettaGenFF at room temperature (300K) are collected.”

$$\Delta S_{torsion} = RT \sum_i \sum_j (-p_{ij} \log p_{ij})$$

where R is the gas constant, T is 300K

In Line 400, “The predicted binding affinity is then calculated using the equation:

$\Delta G = \Delta H - T\Delta S = \Delta H + \Delta S$, where temperature T is implicitly included in ΔS estimation described above.”

- p9. A few more details are needed around which terms are kept and which are removed from RosettaGenFF. Is the binding energy a sum of Coulomb + LJ terms? No mention is made of the solvation term, so has this been removed? If that is the case, what desolvation penalty is present in the scoring function? I would expect optimization in the absence of this term to produce highly polar ligands that would also be highly water soluble.

We apologize for the confusion. The new RosettaGenFF-VS is using the exact RosettaGenFF for ΔH estimation plus a newly developed entropy term for binding affinity estimation. We have revised the “Binding affinity estimation” section in the manuscript :

“RosettaGenFF combines the Rosetta protein energy model⁵⁶⁻⁵⁸ and generic energy terms for non-protein molecules, i.e. Lennard-Jones, Coulomb, hydrogen-bond, implicit-solvation and generic torsion energy. A complete description of RosettaGenFF can be found in Ref²³.”

- p14. Maybe related to the above, is it problematic that only 160 out of 1000 compound pass simple filters? This seems to indicate the post-processing is just as important a factor in compound design as the scoring function. Which criterion removed the most molecules?

We appreciate the reviewer's insightful comment. Yes, the filters are very important to reduce the false positives. This is the reason we integrated the filters into our virtual screening pipeline. Unsatisfied hydrogen bonds and torsion angle outliers filter are the two filters that removed the most molecules. Many of the top scoring compounds failing to pass the filters indicates that there is still a large room for scoring function improvement, which is certainly a future research direction; Our filters suggest that these conformations are "underpenalized" by the current energy model. However, we believe that the scoring function is still the most important factor to enrich the true active compounds in the very top percentage of the ranked compounds in any virtual screening campaign. In fact, it is the superior performance of the scoring function that allowed us to filter only the top 1000 ranked compounds out of the several million docked compounds and still find novel hit compounds.

In the manuscript, we have revised the Methods "**Filters for selecting promising compounds.**" and "**Filtering and clustering**",

"These filters helped us **in reducing the false positives** and refining our selection of compounds for final experimental validation."

For filtering KLHDC2 compounds, we added the following highlighted text:

"We took the top-ranked 1,000 compounds and filtered out compounds with low predicted solubility **(removing 93 compounds)**, unsatisfied hydrogen bonds in the bound conformation **(removing 754 compounds)**, yielding 153 compounds."

and for filtering Nav1.7 compounds, we added the following highlighted text:

"We removed molecules with low predicted solubility **(removing 183 compounds)**, unsatisfied hydrogen bonds in the bound conformation **(removing 139 compounds)** and torsion angle outliers **(removing 520 compounds)** in Cambridge Structural Database (CSD)"

"The filters for unsatisfied hydrogen bonds and torsion angle outliers are the two that removed the most molecules. Although the current energy model achieves state-of-the-art performance, our filters suggest that these conformations are 'underpenalized' by the current energy model, indicating an opportunity for improvement in the scoring function."

- p12. Deployment of the active learning model on the entire enamine library is an impressive calculation. Can the authors please comment on the computational resources and cost of this step in particular? What is the accuracy of the final ML model at the task of predicting docking scores?

We thank the reviewer for the insightful comments. We have mentioned the Active learning model in Methods "*The choice of the FFN was made to reduce the cost of the surrogate model*". With precomputed fingerprints, inferring one million molecules will take around 110 seconds on average using a single cpu (Intel Xeon E5-2695 v3 @ 2.30GHz) or around 11 seconds on average using a RTX2080 gpu. The final ML model is a good classifier based on the AUC on the independent test subset.

We have added the following in the Methods "Active learning model":

“With precomputed fingerprints, the inference of one million molecules using this model will take, on average, around 110 seconds using a single CPU (Intel Xeon E5-2695 v3 @ 2.30GHz), or around 11 seconds on average using an RTX2080 GPU. The model performance was monitored throughout the active learning process and the final model for KLHDC2 has an AUC of 0.886 and the final model for the Nav1.7 target has an AUC of 0.927 on the independent test set, which shows the final ML model is indeed a good binary classifier for each target.”

Reviewer #2 (Remarks on code availability):

I have not installed and run the code. However I have reviewed the supplied data, README files and installation instructions. Everything seems to be thoroughly documented and complete, I just have one comment:

- p10. Please clarify whether the new force field parameters are publicly available.

We thank the reviewer for the positive comment. RosettaGenFF and RosettaGenFF-VS are already in the latest version of Rosetta, which is freely available to academic users but requires a license for commercial users.

Reviewer #3 (Remarks to the Author):

The manuscript titled "An artificial intelligence accelerated virtual screening platform for drug discovery" by Guangfeng Zhou and colleagues introduces a cutting-edge structure-based virtual screening method known as RosettaVS. This study is poised to enhance the efficiency and precision of early-phase drug discovery efforts by accurately predicting the binding poses and affinities of compounds from extensive chemical libraries. The authors have developed an open-source platform, OpenVS, which harnesses the power of artificial intelligence to expedite the virtual screening process. The RosettaVS method shows promise in surpassing current state-of-the-art techniques, with the capacity to manage libraries comprising billions of compounds. The platform's efficacy was demonstrated through its application to two targets: KLHDC2, a novel ubiquitin ligase, and the human voltage-gated sodium channel Nav1.7, yielding several novel hits with single-digit micromolar binding affinities within a remarkably short timeframe of one week.

Despite the innovative nature of the approach to virtual screening and drug discovery, the manuscript raises several concerns regarding the model's innovativeness, precision, and presentation of results. The manuscript does not convincingly establish the novelty and accuracy of the RosettaVS model when benchmarked against existing, well-established methods. To fortify the study's claims, we suggest a comprehensive comparison with current leading-edge techniques, which would yield a more objective evaluation of the model's performance.

We appreciate the reviewer's suggestion. In the manuscript, we have highlighted that RosettaVS uses a state-of-the-art, *physics-based* scoring function, RosettaGenFF-VS. We have observed that the deep learning models may not generalize well to unseen examples, which is why our primary comparison has been with other well-established physics-based scoring functions. However, to provide a more comprehensive evaluation of the model's performance,

we have included several state-of-the-art deep-learning based scoring functions in the SI data. It is noteworthy that RosettaGenFF-VS continues to be one of the top-performing scoring functions, as evidenced by CASF2016 docking and screening power.

We have revised the manuscript to include this information:

“Although our primary comparison is with other well-established physics-based scoring functions in this work, to provide a more comprehensive evaluation of the model’s performance, we have compared our model to several state-of-the-art deep-learning based scoring functions (see Supplementary Data 1), where RosettaGenFF-VS fares favorably.”

The following specific areas require further attention:

The manuscript would be strengthened by the incorporation of permutation tests or other robust statistical methodologies to evaluate the significance of the model's performance. Such methodologies would lend rigor to the interpretation of the findings and bolster the model's claimed effectiveness.

We appreciate the reviewer’s suggestion. In Fig. 1b, 95% confidence intervals were shown as error bars. In Fig 1.e, 90% confidence intervals were computed using the bootstrapping method provided by the CASF2016 package. For Figure clarity, we didn’t include the confidence intervals for CASF2016 docking power (Fig. 1d) and the success rate of screening power (Fig. 1f). Instead, we have included the 90% confidence interval of our method for CASF2016 docking power and screening power in the SI data.

We have revised the Fig 1 caption:

“b, Results of the area-under-curve (AUC) of the receiver operator characteristics (ROC) curve of the DUD benchmark, averaged over targets and averaged from three independent runs, 95% confidence intervals are shown as error bars.; c, Mean ROC enrichments of the DUD benchmark at 0.5% e, CASF2016 screening power, the top 1% enrichment factors with 90% confidence interval of the top ten methods are shown. ... Results of other methods are obtained from Ref²⁸.”

The manuscript should provide a more detailed explanation of the uncertainty estimation methods employed in the active learning component of the OpenVS platform. Transparency regarding these methods is essential for grasping the dependability of the model's predictions and ensuring the reproducibility of the research.

We thank the reviewer for this comment. We apologize for not being clear about the uncertainty estimation in the active learning process, and we fully agree with the reviewer on the importance of ensuring the reproducibility of the research. In this work, we did not use any explicit uncertainty estimation in the active learning process. Instead, we used a greedy strategy for selecting promising compounds. Uncertainty estimation can be obtained using Monte Carlo (MC) dropout by running model inference multiple times with activated dropout layers within the current framework. However, as pointed out by Reviewer #2, using the active learning model, even a very simple one, on the entire multi-billion compound library is associated with expensive calculations. We intentionally chose the greedy strategy to reduce both computational cost and inference time.

We have revised the manuscript to include details about the computational cost of model inference and to provide more clear information on uncertainty estimation.

“With precomputed fingerprints, the inference of one million molecules using this model will take, on average, around 110 seconds using a single CPU (Intel Xeon E5-2695 v3 @ 2.30GHz), or around 11 seconds on average using an RTX2080 GPU.”

“Workflow of AI accelerated virtual screening. *In this work, we utilized active learning techniques to guide the exploration of the vast chemical space. The greedy strategy was used to select new compounds for each iteration to augment the training dataset without using any explicit uncertainty information to reduce the computational cost and inference time. Uncertainty estimation can be obtained using Monte Carlo (MC) dropout⁷⁰ by running model inference multiple times with activated dropout layers within the current framework. The concrete workflow is described as follows.* “

The study could be markedly improved by integrating key drug-likeness metrics, such as logP, QED score, or Synthetic Accessibility score, into the model. These parameters are pivotal for assessing drug-likeness and should be thoroughly examined in the context of the KLHDC2 and Nav1.7 case studies for a holistic evaluation.

We appreciate the reviewer’s constructive suggestion. logP is already computed for the final set of molecules to guide the selection of molecules for experimental validation. Synthetic accessibility score is not necessary in this work since all of the compounds from Enamine REAL or ZINC22 library are synthesizable with around 80% success rate. However, we fully agree with the reviewer that these drug-likeness metrics are important to assess drug-likeness and could easily be added to the pipeline for a comprehensive evaluation of the screened compounds.

We have modified our scripts to report these metrics when collecting results from virtual screening and we also include a new jupyter notebook file to easily report these metrics for a final set of molecules.

We have revised the manuscript and SI to report these properties for the ordered compounds.

“Common chemical properties. *We have reported several chemical properties that are important in small molecule drug discovery, including the quantitative estimate of drug-likeness (QED)⁷⁴, the calculated octanol-water partition coefficient (cLogP)⁷⁵, and synthetic accessibility (SA)⁷⁶. As shown in Table S2, the compounds ordered for both targets have an average QED above 0.7, indicating a high degree of drug-likeness. The cLogP values of the compounds fall within a reasonable range, as it is used as a filter to remove compounds that are excessively hydrophobic (see ‘Filters for selecting promising compounds’). The average SA is low, as expected, because all the compounds from Enamine REAL or ZINC22 are highly synthesizable.* “

Supplementary Table 2 | Chemical properties of ordered compounds.

Target	KLHDC2			Nav1.7		
Property	QED ^a ↑	cLogP ^b	SA ^c ↓	QED ^a ↑	cLogP ^b	SA ^c ↓
min	0.43	-2.16	2.24	0.57	0.80	3.16

mean	0.69	0.83	3.35	0.76	2.40	3.81
max	0.87	2.61	4.88	0.85	3.47	4.35

The statistics of important chemical properties of ordered molecules are shown in this table. ^aQED, Quantitative Estimate of Drug-likeness, ranges from 0 (drug-unlike) to 1 (drug-like)⁷⁴. ^bcLogP, the calculated octanol-water partition coefficient, assesses the hydrophobicity of organic compounds, higher value means more hydrophobic⁷⁵. ^cSA, synthetic accessibility, quantifies the difficulty of chemical synthesis of organic compounds with values between 0 (easy to synthesize) to 10 (difficult to synthesize)⁷⁶.

We strongly advocate for the execution of redocking experiments for the virtual screening of KLHDC2 and Nav1.7. This additional validation step is crucial for confirming the model's predictions and ensuring the reliability of the identified binders.

We appreciate the reviewer for raising this concern. The redocking of all the compounds for both KLHDC2 and Nav1.7 targets will require a large amount of computations. However, to validate the reliability of RosettaVS, we performed a rescreening experiment for the KLHDC2 target using ~4.1 million compounds from the Enamine Screening Library, which contains the first samples of billions of compounds that can be synthesized. Our previous best hit C2.8 was included in this screening library. The best hit compound C2.8 was rediscovered and ranked among the top 1000 compounds (within 0.024% of the 4.1 million compounds). After we applied the buried unsatisfied hydrogen bond filter to remove any docked complex that had more than one buried polar atom, the best hit C2.8 was among the top 50 compounds. We believe this experiment further confirms the effectiveness and reliability of our methods to identify binders.

We have revised the manuscript to include this new experiment:

Line 248: *"To test the reliability of our screening procedure, we reran a portion of our computational experiment and rediscovered the confirmed best hit compound C2.8. (see Methods for details)"*

In Methods, we added: *"Rescreening against KLHDC2. To further validate the reliability of RosettaVS for hit discovery, we performed a rescreening experiment against the KLHDC2 target using the Enamine Screening library. This library contains around 4.1 million compounds, which are the first samples of billions of synthesizable compounds. Our confirmed best hit compound, C2.8, was included in this screening library. We were able to rediscover compound C2.8 from this experiment. In fact, it was among the top 1000 compounds, which is within 0.024% of the 4.1 million compounds, before any filtering. After applying the buried unsatisfied hydrogen bond filter, we removed any docked complex that had more than one buried polar atom. Following this, the hit compound C2.8 was among the top 50 compounds. The success of this rescreening experiment shows our virtual screening protocol is both effective and reliable for hit discovery."*

Lastly, the manuscript's figure presentation requires refinement. The figures must conform to the journal's guidelines for clarity, resolution, and informative labeling. It is imperative to ensure that all visual components are of the highest quality and accurately depict the data under communication.

We apologize for this; the figures for initial review were uploaded in low resolution. We have uploaded higher-resolution figures in this resubmission.

REVIEWERS' COMMENTS

Reviewer #1 (Remarks to the Author):

I would recommend this nice work be published without further modification.

Reviewer #2 (Remarks to the Author):

The authors have fully addressed all of my comments from the first round of review, and I recommend publication.

Reviewer #2 (Remarks on code availability):

See my first report.

Reviewer #3 (Remarks to the Author):

The authors have adequately addressed all my concerns.